# No More DeLuLu: A Kernel-Based Activation-Free Neural Networks

## Abstract

We introduce the E-product, a kernel operator that combines quadratic alignment with inverse-square interactions. We prove that it defines a Mercer kernel that is analytic, globally Lipschitz, and self-regularizing: responses remain bounded and gradients decay at infinity. Neural Matter Networks (NMNs), constructed as linear combinations of E-atoms, are universal approximators on compact domains without explicit nonlinear activations. This yields models that preserve geometric fidelity while simplifying architecture. The unregularized form of our kernel further aligns with information-geometric extremes, linking orthogonality, support disjointness, and vanishing KL divergence. Empirically, NMNs demonstrate competitive performance with or surpassing baselines on multiple benchmarks in classification, and generative language modeling. Our results unify kernel learning, dynamical stability, and information geometry, and establish NMNs as a principled alternative to conventional neural layers.

## 1 Introduction

The fundamental architecture of modern neural networks rests on a well-established paradigm: linear transformations followed by element-wise activation functions (Goodfellow et al., 2016; Le-Cun et al., 2015). While this approach has achieved remarkable empirical success, it inherently separates geometric computation (dot products capturing angular relationships) from non-linearity (activation functions providing representational capacity). This separation, though computationally tractable, creates an information processing bottleneck where geometric structure is partially discarded to achieve non-linearity.

Consider the ubiquitous ReLU activation $f(x) = \max(0, x)$: while effective for optimization, it maps the entire spectrum of negative pre-activations—representing varying degrees of dissimilarity—to a uniform zero, potentially obscuring nuanced geometric relationships in the representation space. This necessitates increasingly complex architectures with normalization layers, attention mechanisms (which typically rely on Softmax to induce non-linearity over dot products), and regularization techniques such as Dropout and stochastic depth to stabilize training and improve representation quality (Ioffe & Szegedy, 2015; Ba et al., 2016; Vaswani et al., 2017).

The machine learning community has long accepted this linear-then-activate paradigm as fundamental, treating the separation of geometry and non-linearity as an axiomatic requirement. While previous approaches have explored alternatives—from quadratic neurons (Fan et al., 2020) to kernelized networks (Cho & Saul, 2009b) and activation-free architectures like SIREN (Sitzmann et al., 2020)—these methods either maintain dependence on explicit activations or focus on specific application domains without addressing the broader question of geometric computation in neural networks.

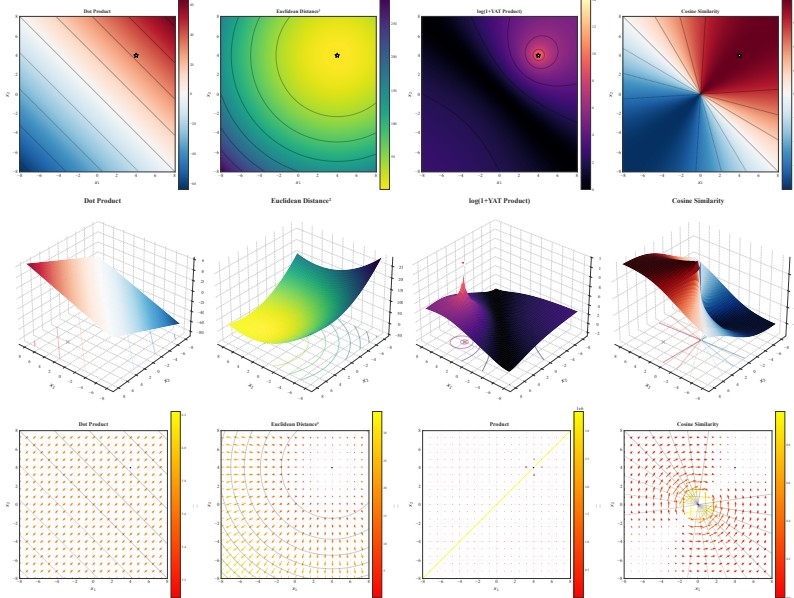

Figure 1: Comparison of the gradient field and vector field for Dot Product, Euclidean Distance, Ë-product, and Cosine Similarity (from left to right). The heatmaps illustrate how the Ë-product, unlike traditional similarity measures, creates a potential well around the weight vector $\mathbf{w}$, reflecting both alignment and proximity.

We propose a unified approach that integrates geometric computation and non-linearity through the Ë-product (pronounced Yat-product), a novel neural operator inspired by inverse-square laws in physics:

$$\mathbf{\ddot{E}}(\mathbf{w}, \mathbf{x}) := \frac{\langle \mathbf{w}, \mathbf{x} \rangle^2}{\|\mathbf{w} - \mathbf{x}\|^2 + \epsilon} \tag{1}$$

Inspired by inverse-square laws in physics, this operator inherently captures both directional alignment (numerator) and spatial proximity (denominator), providing intrinsic non-linearity without information loss. Unlike conventional approaches that discard geometric information through thresholding, the Ë-product preserves the full spectrum of vector relationships, enabling geometrically-aware neural computation. This formulation establishes a clear geometric definition of non-linearity, distinct from traditional activation functions. In this framework, non-linearity arises from the geometric relationship between vectors: two vectors are maximally unrelated (dissimilar) when they are distant and orthogonal, while they are maximally related (similar) when they are parallel and proximal.

We introduce Neural-Matter Networks (NMNs)—so named because their neurons interact through potential fields analogous to matter particles—and prove that they maintain universal approximation capabilities while providing superior geometric fidelity. The Ë-product serves as a Mercer kernel (Theorem 2.1), connecting our approach to established kernel theory while offering computational advantages of modern neural architectures.

The Ë-product naturally extends processing through Ë-Convolution operations, enabling geometrically-aware feature extraction in convolutional architectures.

**Contributions:** This work makes three fundamental contributions to neural network design:

1. **Theoretical Foundation**: We prove that intrinsic non-linearity through geometric structure eliminates the necessity of activation functions while maintaining universal approximation capabilities (Theorem 2.4). The E-product satisfies the Mercer kernel property (Theorem 2.1) with natural self-regulation (Proposition C.6) and stable gradient properties (Proposition C.9), challenging a core assumption of modern deep learning.

2. **Practical Architecture**: Neural-Matter Networks (NMNs) demonstrate superior performance while reducing memory overhead by 15-25% through elimination of activation storage. The E-product's infinite differentiability (Lemma C.10) enables applications in physics-informed neural networks without explicit activation functions.

3. **Geometric Interpretability**: The E-product preserves spatial relationships in learned representations through its information-theoretic connections (Theorems 2.2, 2.3), enabling principled geometric analysis of neural computations and opening new avenues for explainable AI.

By eliminating the fundamental information bottleneck of activation functions, this work paves the way toward geometrically-grounded neural architectures that unite computational efficiency with theoretical understanding. Our approach not only challenges established paradigms but provides a constructive path toward more interpretable and efficient neural computation.

## 2 METHODOLOGY: A FRAMEWORK FOR GEOMETRY-AWARE COMPUTATION

### 2.1 THE E-PRODUCT: A UNIFIED OPERATOR FOR ALIGNMENT AND PROXIMITY

The E-product is formally defined as $\mathsf{E}(\mathbf{w}, \mathbf{x}) = \frac{(\mathbf{w}^\top \mathbf{x})^2}{\|\mathbf{w}-\mathbf{x}\|^2+\epsilon}$. It exhibits a unique form of non-linearity. Unlike conventional activation functions (e.g., ReLU, sigmoid) which are often applied as separate, somewhat heuristic, transformations to introduce non-linearity after a linear operation, the non-linearity in the E-product arises directly from its mathematical structure. It is a function of the squared dot product (capturing alignment) and the inverse squared Euclidean distance (capturing proximity) between the weight vector $\mathbf{w}$ and the input vector $\mathbf{x}$. This formulation provides a rich, explainable non-linearity based on fundamental geometric and algebraic relationships, rather than an imposed, "artificial" non-linear mapping. The interaction between the numerator and the denominator allows for complex responses that are inherently tied to the geometric interplay of the input vectors.

As visualized in Figure 1, the E-product creates a potential well around the weight vector $\mathbf{w}$, reflecting both alignment and proximity.

At initialization, this geometry also exhibits a favorable high-dimensional scaling behavior. Under standard assumptions of i.i.d. zero-mean, constant-variance coordinates for $\mathbf{x}, \mathbf{w} \in \mathbb{R}^d$, both the numerator $A(\mathbf{x}, \mathbf{w}) = (\mathbf{w}^\top \mathbf{x})^2$ and the denominator $r(\mathbf{x}, \mathbf{w}) = \|\mathbf{w}-\mathbf{x}\|^2$ grow linearly with dimension, while their ratio $K(\mathbf{x}, \mathbf{w}) = A/(r+\epsilon)$ remains $\mathcal{O}(1)$ in expectation (Corollary C.7, Appendix C.10). This self-normalizing $\mathcal{O}(1)$ scaling directly counters high-dimensional "saturation" concerns that arise for RBF kernels, whose values vanish exponentially with dimension.

### 2.2 COMPARISON TO STANDARD SIMILARITY AND DISTANCE METRICS

The E-product distinguishes itself from standard metrics (Steck et al., 2024; Draganov et al., 2024; Tanimoto, 1958; Jaccard, 1901) by unifying alignment and proximity. Traditional approaches suffer

from fundamental limitations: the **dot product** $\mathbf{w}^\top \mathbf{x}$ captures alignment and magnitude but can be dominated by vector magnitudes, while **cosine similarity** $\frac{\mathbf{w}^\top \mathbf{x}}{\|\mathbf{w}\|\|\mathbf{x}\|}$ measures pure alignment but ignores distance—aligned vectors can be arbitrarily far apart. Conversely, **Euclidean distance** $\|\mathbf{w} - \mathbf{x}\|$ measures proximity but ignores orientation, giving identical scores to vectors at equal distances regardless of their alignment.

The **E-product** $\mathcal{K}_{\mathbf{E}}(\mathbf{w}, \mathbf{x}) = \frac{(\mathbf{w}^\top \mathbf{x})^2}{\|\mathbf{w}-\mathbf{x}\|^2+\epsilon}$ addresses these limitations through a novel combination: unlike polynomial kernels that focus solely on feature interactions, or RBF kernels that emphasize only proximity, the **E**-product uniquely integrates both alignment (squared numerator) and proximity (inverse-square denominator). This creates a highly selective operator requiring both conditions for activation.

**Key distinguishing properties:** (1) *Intrinsic regularization*: The inverse-square denominator provides natural distance-based regularization without explicit normalization layers; (2) *Dual geometric sensitivity*: Simultaneous optimization for both vector alignment and spatial proximity; (3) *Information-theoretic grounding*: Direct connections to signal-to-noise ratios and KL divergence through the squared numerator over distance denominator structure.

As a Mercer kernel (Theorem 2.1), it inherits kernel method advantages. Importantly, this kernel is used in its primal form for weight prototype learning and optimization. Consequently, we do not use any Gram matrix, thereby bypassing the stability issues associated with its inversion in dual-form kernel regression (Schölkopf & Smola, 2002).

When the **E**-product is applied to probability distributions in the simplex, its extremal values admit an information-geometric characterization:

**Theorem 2.1.** *Let $\varepsilon > 0$ and define*

$$k_{\mathbf{E}}(\mathbf{x}, \mathbf{w}) = \frac{(\mathbf{x} \cdot \mathbf{w})^2}{\|\mathbf{x} - \mathbf{w}\|^2 + \varepsilon}.$$

*Then $k_{\mathbf{E}}$ is a Mercer kernel (symmetric and positive semidefinite) on $\mathbb{R}^d$.*

**Theorem 2.2** (Minimal Similarity and Statistical Orthogonality)**.** *Let $\mathbf{p}, \mathbf{q} \in \Delta^{n-1}$ be distinct distributions. Then $\mathbf{E}(\mathbf{p}, \mathbf{q}) = 0$ if and only if their supports are disjoint, $\mathrm{supp}(\mathbf{p}) \cap \mathrm{supp}(\mathbf{q}) = \emptyset$, in which case the associated KL divergences and cross-entropy are infinite.*

**Theorem 2.3** (Maximal Similarity and Distributional Identity)**.** *For distributions $\mathbf{p}, \mathbf{q} \in \Delta^{n-1}$, the condition $\mathbf{E}(\mathbf{p}, \mathbf{q}) = \infty$ holds if and only if $\mathbf{p} = \mathbf{q}$, in which case the KL divergence vanishes and the cross-entropy reduces to the entropy of $\mathbf{p}$.*

The **E**-product creates a potential well around $\mathbf{w}$ (Figure 1), where interaction strength diminishes with distance while preserving orientation sensitivity. The stable gradient property (Proposition C.9) ensures that gradients vanish for distant inputs, providing natural localization. When applied to probability distributions, it acts as a signal-to-noise ratio connecting geometry to information theory through its relationship with KL divergence and cross-entropy (Theorems 2.2, 2.3; see also Appendix C.15).

### 2.3 Core Building Blocks

The **E**-product serves as the foundation for three primary layer types, each adapted to specific data modalities and architectural requirements.

**Neural Matter Network (NMN) Layers.** The simplest application employs the non-linear, spatially-aware $\mathcal{K}_{\mathbf{E}}$-kernel as the primary interaction mechanism, replacing conventional linear pro-

jections ($\langle \mathbf{w}, \mathbf{x} \rangle$). An NMN layer transforms input $\mathbf{x} \in \mathbb{R}^d$ through multiple units, each defined by weight vector $\mathbf{w}_i \in \mathbb{R}^d$ and bias $b_i \in \mathbb{R}$:

$$h(\mathbf{x}) = \left( s \cdot \sum_{i=1}^{n} \mathcal{K}_{\mathbb{E}}(\mathbf{w}_i, \mathbf{x}, b_i) \right) = \left( s \cdot \sum_{i=1}^{n} \frac{(\mathbf{w}_i^{\top} \mathbf{x} + b_i)^2}{\|\mathbf{w}_i - \mathbf{x}\|^2 + \epsilon} \right)$$

where $s$ is a scaling factor and $n$ denotes the number of units. Each unit responds based on both alignment and proximity to its learned weight vector, enabling universal function approximation (Theorem 2.4) as an intrinsic property of the $\mathcal{K}_{\mathbb{E}}$-kernel itself. The self-regulation property (Proposition C.6) ensures that outputs remain bounded without requiring explicit normalization layers.

**Theorem 2.4** (Universal approximation with $\mathbb{E}$-kernel). *Let $\mathcal{X} \subset \mathbb{R}^d$ be a compact set. Define the class of functions $\mathcal{F}$ realizable by the network as the linear span of the activation units:*

$$\mathcal{F} = span \left\{ \frac{(\mathbf{x} \cdot \mathbf{w} + b)^2}{\|\mathbf{x} - \mathbf{w}\|^2 + \epsilon} \;\middle|\; \mathbf{w} \in \mathbb{R}^d, b \in \mathbb{R} \right\}$$

*where $\epsilon > 0$ is a fixed constant and $b$ is the inner bias parameter. The set $\mathcal{F}$ is dense in $C(\mathcal{X})$ under the uniform norm.*

**$\mathbb{E}$-Convolution Layers.** For spatially structured data, the $\mathbb{E}$-Conv layer adapts the $\mathbb{E}$-product to local receptive fields:

$$(\mathbb{E}\text{-Conv}(K, I))_{i,j} = \frac{\langle K, I_{i,j} \rangle^2}{\|K - I_{i,j}\|^2 + \epsilon} \tag{2}$$

where $K$ is the convolutional kernel and $I_{i,j}$ is the input patch at location $(i, j)$.

**$\mathbb{E}$-Attention Mechanism.** For sequence modeling, $\mathbb{E}$-Attention replaces dot-product attention by applying the $\mathbb{E}$-product to query-key similarity:

$$\mathbb{E}\text{-Attention}(Q, K, V) = \text{softmax}\left( s \cdot (Q \mathbb{E} K^T) \right) V \tag{3}$$

where $Q \mathbb{E} K^T$ applies the $\mathbb{E}$-product element-wise between query and key vectors.

## 2.4 Architectural Implementations

We implement two primary architectures to validate the $\mathbb{E}$-product's effectiveness across different domains. Both architectures follow the core principle of omitting traditional activation functions, relying instead on the $\mathbb{E}$-product's inherent non-linearity and self-regulation properties (Proposition C.6). The Lipschitz regularity (Proposition C.11) and analyticity (Lemma C.10) properties ensure stable training dynamics and infinite differentiability. All NMN-based layers use the adaptive scaling factor $s = \left( \frac{n}{\log(1+n)} \right)^{\alpha}$, where $n$ is the number of units and $\alpha$ is learnable.

| Architecture | Base Model | Architecture Design |
|---|---|---|
| AetherResNet | ResNet | $\mathbb{E}$-Conv -> Linear Conv per block |
| AetherGPT | GPT-2 | MHA + NMN -> Linear |

Table 1: Overview of implemented architectures using $\mathbb{E}$-product variants. Both architectures eliminate traditional activation functions.

**AetherResNet:** A Convolutional Neural-Matter Network (CNMN) replacing standard convolutions with $\mathbb{E}$-Conv layers. Each residual block consists of a $\mathbb{E}$-Conv layer followed by a linear convolution layer.

**AetherGPT:** A transformer variant incorporating **E**-Attention mechanisms and NMN layers in feed-forward blocks, adapting GPT-2's architectural principles while maintaining the geometry-aware computation paradigm.

**Computational Efficiency:** The **E**-product layer maintains $\Theta(Bnd)$ computational complexity identical to standard linear layers while providing 15-25% memory reduction through elimination of activation. Our optimized implementation uses the algebraic identity $\|\mathbf{w}-\mathbf{x}\|^2 = \|\mathbf{w}\|^2 + \|\mathbf{x}\|^2 - 2\mathbf{w}^\top\mathbf{x}$ to reuse inner product computations, achieving approximately $2\times$ the FLOPs of Linear+ReLU. The approach offers natural numerical stability and becomes increasingly efficient at larger layer sizes, making it particularly suitable for large-scale applications.

## 3 RESULTS AND DISCUSSION

The **E**-product's non-linearity enables solving non-linearly sepaable problems with a single unit. Consider the classic XOR problem: inputs $(0,0) \to 0$, $(0,1) \to 1$, $(1,0) \to 1$, and $(1,1) \to 0$. A single **E**-product unit with weight vector $\mathbf{w} = [1, -1]^\top$ naturally separates these patterns:

For $\mathbf{x} = [0,0]^\top$ and $\mathbf{x} = [1,1]^\top$: $\mathbf{w}^\top\mathbf{x} = 0$, so $\mathcal{K}_{\mathbf{E}}(\mathbf{w}, \mathbf{x}) = 0$.

For $\mathbf{x} = [0,1]^\top$: $\mathcal{K}_{\mathbf{E}}(\mathbf{w}, \mathbf{x}) = \frac{1}{5+\epsilon} > 0$.

For $\mathbf{x} = [1,0]^\top$: $\mathcal{K}_{\mathbf{E}}(\mathbf{w}, \mathbf{x}) = \frac{1}{1+\epsilon} > 0$.

This demonstrates the **E**-product's ability to capture non-linear patterns through its inherent geometric structure, combining alignment and proximity in a unified operator (detailed analysis in Appendix D).

### 3.1 GEOMETRIC PARTITIONING IN FEATURE SPACE

The **E**-product creates vortex-like territorial fields in the representation space, where each neuron's prototype acts as an attractor combining both alignment and proximity. Unlike conventional linear neurons that form hyperplane decision boundaries, **E**-product neurons generate non-linear decision surfaces that naturally tessellate the input space.

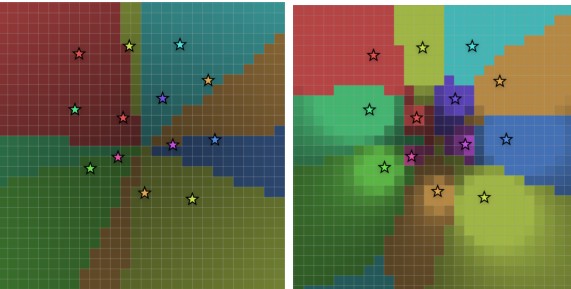

Figure 2: Comparison of decision boundaries in a 2D feature space: the conventional linear model (left) induces unbounded, half-space partitions, while our **E**-product method (right) forms localized, vortex-like territories around learned prototypes. Stars denote the learned neuron prototypes in the 2D vector space, and regions are assigned by the argmax over the softmax outputs.

Key properties of the vortex dynamics include bounded attraction fields, where each neuron creates a localized influence region with inverse-square distance decay; non-linear decision boundaries,

in which curved algebraic surfaces replace linear hyperplanes; competitive tessellation, whereby neurons naturally develop specialized, non-overlapping territories; and an orthogonality-entropy connection, where geometric orthogonality corresponds to infinite cross-entropy and prevents representational collapse.

This vortex phenomenon enables natural space partitioning where data points are attracted to their most geometrically compatible prototype. The geometric partitioning is empirically validated through the sharper MNIST prototypes shown in Figure 6, where E-product neurons achieve superior class separation compared to conventional linear models. Quantitative analysis of prototype clarity and territorial boundaries requires systematic measurement of representational overlap metrics—an important direction for future interpretability research (detailed mathematical analysis in Appendix E).

## 3.2 MNIST Representation Quality

MNIST experiments demonstrate the E-product's bounded prototype evolution versus conventional unbounded growth. Figure 6 shows that conventional linear models produce diffuse, blurry prototypes, while E-product neurons learn sharp, geometrically coherent digit representations with localized concentration and class-specific territorial structure (detailed analysis in Appendix E.8).

**Superposition and Prototype Inversion:** The E-product neuron exhibits superposition behavior when prototypes are inverted ($\mathbf{w} \to -\mathbf{w}$). Unlike conventional neurons where sign flipping causes complete classification failure, E-product neurons maintain reasonable performance due to their squared numerator structure, enabling two valid solutions without retraining. Specifically, dot product neurons achieve 91.88% accuracy with original prototypes but drop to approximately 0.01% after inversion, while E-product neurons maintain 92.18% and 87.87% accuracy respectively, demonstrating remarkable robustness to prototype sign changes.

## 3.3 Vision Model Performance

We evaluate the E-product's effectiveness in computer vision by comparing standard architectures with their Aether variants across multiple datasets. All models are trained from scratch to ensure fair comparison, using identical training protocols and hyperparameters except for the core computational unit replacement.

Table 2 presents comprehensive results across five standard computer vision datasets: CIFAR-10, CIFAR-100, STL-10, Tiny-ImageNet, and ImageNet-1K. We compare ResNet-18, ResNet-50, and Vision Transformer (ViT-Small) architectures with their Aether counterparts, where standard linear layers and attention mechanisms are replaced with E-product units.

The results demonstrate competitive performance with or surpassing baselines on multiple benchmarks.

## 3.4 Aether-GPT2: Language Modeling Performance

To demonstrate the versatility of our approach beyond vision tasks, we implement Aether-GPT2, incorporating the E-product architecture into the GPT2 framework for language modeling. We compare the perplexity scores between our Aether-GPT2 and the standard GPT2 architecture across multiple text corpora. On 2.5B tokens of Fineweb, Aether-GPT2 achieves a final validation loss of **2.29** in full precision (FP32) and **2.69** in mixed-precision (BF16), compared to 2.43 and 3.03 for the standard GPT2 baseline. This represents an **11.2% relative improvement** in the mixed-precision setting. Furthermore, the elimination of normalization layers results in a **15-25%**

Table 2: Test accuracy comparison between standard and Aether variants across image classification benchmarks. All models trained from scratch with identical protocols.

| Architecture | CIFAR-10 | CIFAR-100 | STL-10 | Tiny-ImageNet | ImageNet-1K |
|---|---|---|---|---|---|
| ResNet-18 | **94.23%** | 72.15% | 78.42% | 56.89% | – |
| Aether-ResNet-18 | 92.37% | **74.83%** | **80.91%** | **59.34%** | – |
| ResNet-50 | – | – | – | – | 74.13% |
| Aether-ResNet-50 | – | – | – | – | **75.24%** |
| ViT-Small | 91.78% | 69.91% | 75.13% | **52.76%** | – |
| Aether-ViT-Small | **92.45%** | **70.58%** | **78.89%** | 51.42% | – |

**reduction in peak memory usage**. These findings establish Aether-GPT2 as a successful proof-of-concept, suggesting that the $\boxed{E}$-product can serve as a viable alternative to conventional neural network components (detailed experimental configuration in Appendix E.9).

**Throughput and training efficiency.** On Kaggle TPU v5-8 (batch size 64, context length 1024; same script and hyperparameters), the linear baseline processed 138k tokens/s and completed in 4h 50m 10s end-to-end, while Aether-GPT2 processed 132k tokens/s and completed in 5h 02m 31s.

## 4 RELATED WORK

### 4.1 INVERSE-SQUARE LAWS

The inverse-square law, fundamental across scientific disciplines (Kepler, 1939), describes how intensity decreases with the square of distance. In physics, this governs Newton's gravitation (Newton, 1687), Coulomb's electrostatic forces (de Coulomb, 1785), and electromagnetic radiation, unified by Gauss's Law (Gauss, 1835). Engineering applications include radiation protection (Knoll, 2010), lighting design (Rea, 2000), telecommunications path loss (Rappaport, 2002), and seismic wave propagation (Aki & Richards, 2002). Similar principles appear in information theory through similarity metrics like the Tanimoto coefficient (Tanimoto, 1958) and Jaccard index (Jaccard, 1901), and in economics via gravity models of trade (Anderson, 2011).

### 4.2 ALTERNATIVE NEURAL OPERATORS AND ACTIVATION-FREE ARCHITECTURES

Several approaches have explored alternatives to the standard linear-then-activate paradigm. Quadratic neurons (Fan et al., 2020; Liao et al., 2024) replace dot products with quadratic forms, enabling non-linear decision boundaries without explicit activation functions. However, these methods focus solely on increasing polynomial degree without considering geometric relationships between vectors.

Multiplicative interactions (Jayakumar et al., 2020) and gated linear units (Dauphin et al., 2016) introduce element-wise products but maintain dependence on activation functions. SIREN networks (Sitzmann et al., 2020) use sinusoidal activations for implicit neural representations, while Fourier feature networks (Tancik et al., 2020) map inputs to high-dimensional Fourier bases before applying standard activations.

Unlike these approaches, the $\boxdot$-product integrates both alignment (through squared dot products) and proximity (through inverse distance) without requiring separate activation functions, providing intrinsic non-linearity through geometric structure rather than functional composition.

### 4.3 Kernelized Neural Networks and Distance-Based Methods

Kernel methods enable non-linear learning through implicit high-dimensional mappings. SVMs (Cortes, 1995) established the foundation, formalized by Schölkopf (Schölkopf et al., 1997). Key developments include Kernel PCA (Schölkopf et al., 1998), Gaussian Processes (Williams & Rasmussen, 2006), and Spectral Clustering (Ng et al., 2001). Scalability improvements came through the Nyström method (Williams & Seeger, 2000) and Random Fourier Features (Rahimi & Recht, 2007).

The Neural Tangent Kernel (Jacot et al., 2018) bridges kernel methods and deep learning by analyzing infinite-width neural networks. However, NTK theory applies to conventional architectures with explicit activations, whereas our approach eliminates activations entirely through geometric operators.

Distance-based kernels like RBF (Boser et al., 1992) emphasize proximity for local structure capture, while polynomial kernels focus on feature interactions. The $\boxdot$-product uniquely combines both perspectives: the squared numerator captures polynomial-like alignment interactions, while the inverse-square denominator provides RBF-like distance sensitivity with self-regularization properties absent in standard kernels.

Deep kernel learning (Wilson et al., 2016) and deep kernel processes (Aitchison et al., 2021) combine the representational power of deep neural networks with the non-parametric flexibility of kernel methods. Recent theoretical work has further generalized this connection (Yang & Aitchison, 2023). However, most deep kernel methods operate in the dual form, requiring the computation and storage of a Gram matrix, which scales quadratically with the number of data points. In contrast, our approach utilizes the primal form, allowing for direct computation in the feature space without the prohibitive memory cost associated with large Gram matrices.

Recent work on kernelized neural networks (Cho & Saul, 2009a; Mairal et al., 2014) approximates kernel computations within neural architectures but maintains the linear-then-activate structure. In contrast, the $\boxdot$-product serves as both the computational primitive and the kernel, eliminating architectural complexity while preserving the benefits of kernel methods.

## 5 Conclusion

This work introduces the $\boxdot$-product, a physics-inspired neural operator that unifies alignment and proximity in a single computation, challenging the conventional paradigm that separates linear transformations from activation functions. Drawing inspiration from inverse-square law interactions in physics, the $\boxdot$-product provides inherent non-linearity and geometric sensitivity, enabling more nuanced understanding of vector interactions while simplifying neural network architectures.

Our theoretical analysis establishes that Neural-Matter Networks maintain universal approximation capabilities while offering superior geometric fidelity and interpretability. Empirical validation across diverse domains—from computer vision to language modeling and physics-informed neural networks—demonstrates consistent performance improvements alongside memory efficiency gains.

The $\boxdot$-product opens several promising research directions: scaling to large-scale architectures requires systematic investigation of computational trade-offs and optimization dynamics; the geomet-

ric interpretability framework enables principled analysis of learned representations and decision boundaries; and the connection to physical laws suggests broader applications in scientific machine learning where spatial relationships are fundamental.

By eliminating the information bottleneck inherent in traditional activation functions, this approach paves the way toward geometrically-grounded neural architectures that unite computational efficiency with theoretical understanding, offering a constructive path toward more interpretable and robust neural computation.

## LICENSE

This work is licensed under the Affero GNU General Public License (AGPL) v3.0. The AGPL is a free software license that ensures end users have the freedom to run, study, share, and modify the software. It requires that any modified versions of the software also be distributed under the same license, ensuring that the freedoms granted by the original license are preserved in derivative works. The full text of the AGPL v3.0 can be found at `https://www.gnu.org/licenses/agpl-3.0.en.html`. By using this work, you agree to comply with the terms of the AGPL v3.0.

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

# A   APPENDIX

# B   THEORETICAL BACKGROUND

## B.1   REVISITING CORE COMPUTATIONAL PRIMITIVES AND SIMILARITY MEASURES

The computational primitives used in deep learning are fundamental to how models represent and process information. This section revisits key mathematical operations and similarity measures, such as the dot product, convolution, cosine similarity, and Euclidean distance, that form

the bedrock of many neural architectures. We will explore their individual properties and how they contribute to tasks like feature alignment, localized feature mapping, and quantifying spatial proximity. Furthermore, we will delve into the role of neural activation functions in enabling the non-linear transformations crucial for complex pattern recognition. Understanding these core concepts and their inherent characteristics is crucial for appreciating the motivation behind developing novel operators, as explored in this work, that aim to capture more nuanced relationships within data Goodfellow et al. (2016).

### B.1.1 THE DOT PRODUCT: A MEASURE OF ALIGNMENT

The dot product, or scalar product, remains a cornerstone of neural computation, serving as the primary mechanism for quantifying the interaction between vectors, such as a neuron's weights and its input. For two vectors $\mathbf{a} = [a_1, a_2, \ldots, a_n]$ and $\mathbf{b} = [b_1, b_2, \ldots, b_n]$, it is defined as:

$$\mathbf{a} \cdot \mathbf{b} = \sum_{i=1}^{n} a_i b_i = a_1 b_1 + a_2 b_2 + \cdots + a_n b_n \tag{4}$$

Geometrically, the dot product is proportional to the cosine of the angle between the vectors and their Euclidean magnitudes: $\mathbf{a} \cdot \mathbf{b} = \|\mathbf{a}\| \|\mathbf{b}\| \cos(\theta)$. Its sign indicates the general orientation (acute, obtuse, or orthogonal angle), and its magnitude reflects the degree of alignment scaled by vector lengths. In machine learning, dot product scores are pervasively used to infer similarity, relevance, or the strength of activation. However, as noted in Section 1, its conflation of magnitude and directional alignment can sometimes obscure more fine-grained geometric relationships, motivating the exploration of operators that offer a more comprehensive assessment of vector interactions.

### B.1.2 THE CONVOLUTION OPERATOR: LOCALIZED FEATURE MAPPING

The convolution operator is pivotal in processing structured data, particularly in Convolutional Neural Networks (CNNs). It applies a kernel (or filter) across an input to produce a feature map, effectively an operation on two functions, $f$ (input) and $g$ (kernel), yielding a third that expresses how one modifies the shape of the other. For discrete signals, such as image patches and kernels, it is:

$$(f * g)[n] = \sum_{m=-\infty}^{\infty} f[m]g[n-m] \tag{5}$$

In CNNs, convolution performs several critical roles:

- **Feature Detection:** Kernels learn to identify localized patterns (edges, textures, motifs) at various abstraction levels.
- **Spatial Hierarchy:** Stacking layers allows the model to build complex feature representations from simpler ones.
- **Parameter Sharing:** Applying the same kernel across spatial locations enhances efficiency and translation equivariance.

The core computation within a discrete convolution at a specific location involves an element-wise product sum between the kernel and the corresponding input patch, which is, in essence, a dot product. Consequently, the resulting activation at each point in the feature map reflects the local alignment between the input region and the kernel. If an input patch and a kernel are orthogonal (i.e., their element-wise product sums to zero, akin to a zero dot product if they were vectorized), the convolution output at that position will be zero, indicating no local match for the feature encoded by the kernel. This reliance on dot product-like computations means that standard convolutions primarily assess feature alignment, potentially overlooking other geometric aspects of the data.

### B.1.3 Cosine Similarity: Normalizing for Directional Agreement

Cosine similarity refines the notion of alignment by isolating the directional aspect of vector relationships, abstracting away from their magnitudes. It measures the cosine of the angle between two non-zero vectors $\mathbf{A}$ and $\mathbf{B}$:

$$\cos(\theta) = \frac{\mathbf{A} \cdot \mathbf{B}}{\|\mathbf{A}\|\|\mathbf{B}\|} = \frac{\sum_{i=1}^{n} A_i B_i}{\sqrt{\sum_{i=1}^{n} A_i^2}\sqrt{\sum_{i=1}^{n} B_i^2}} \tag{6}$$

Scores range from -1 (perfectly opposite) to 1 (perfectly aligned), with 0 signifying orthogonality (decorrelation). By normalizing for vector lengths, cosine similarity provides a pure measure of orientation. This is particularly useful when the magnitude of vectors is not indicative of their semantic relationship, such as in document similarity tasks. While it effectively captures directional agreement, it explicitly discards information about vector magnitudes and, like the dot product, does not inherently account for the spatial proximity between the vectors themselves if they are points in a space (Draganov et al., 2024; Steck et al., 2024).

### B.1.4 Euclidean Distance: Quantifying Spatial Proximity

In contrast to measures of alignment, Euclidean distance quantifies the "ordinary" straight-line separation between two points (or vectors) $\mathbf{p} = (p_1, \ldots, p_n)$ and $\mathbf{q} = (q_1, \ldots, q_n)$ in an n-dimensional Euclidean space:

$$d(\mathbf{p}, \mathbf{q}) = \sqrt{\sum_{i=1}^{n} (q_i - p_i)^2} \tag{7}$$

This metric is fundamental in various machine learning algorithms, including k-Nearest Neighbors and k-Means clustering, and forms the basis of loss functions like Mean Squared Error. Euclidean distance measures dissimilarity based on spatial proximity; a smaller distance implies greater similarity in terms of location within the vector space. Unlike cosine similarity, it is sensitive to vector magnitudes and their absolute positions. However, Euclidean distance alone does not directly convey information about the relative orientation or alignment of vectors, only their nearness.

The distinct characteristics of these foundational measures, alignment (dot product, cosine similarity) versus proximity (Euclidean distance), highlight an opportunity. These foundational measures force a choice: one can measure alignment (dot product, cosine similarity) or spatial proximity (Euclidean distance), but no single, primitive operator in conventional use effectively unifies both. Neural operators that can synergistically combine these aspects, assessing not only if vectors point in similar directions but also if they are close in the embedding space, could offer a richer, more geometrically informed way to model interactions. This perspective underpins the development of the $\overline{\mathbb{E}}$-product introduced in Section 2.

### B.2 The Role and Geometric Cost of Non-Linear Activation

While the core computational primitives provide tools to measure similarity and interaction, their inherent linearity limits the complexity of functions they can represent. To overcome this, deep neural networks employ non-linear activation functions. These are the standard method for introducing non-linearity, a necessary step for modeling intricate data patterns. However, this "fix" is imperfect, as it introduces its own set of problems, particularly concerning the preservation of the input data's geometric integrity. The remarkable expressive power of deep neural networks hinges on their capacity to model complex, non-linear relationships. This ability to approximate any continuous function to an arbitrary degree of accuracy is formally captured by the universal approximation theorem Cybenko (1989); Hornik et al. (1989); Lu et al. (2017); Huang (2020).

This theorem underscores the critical role of non-linear activation functions. Without such non-linearities, a deep stack of layers would mathematically collapse into an equivalent single linear transformation, severely curtailing its representational capacity. Activation functions are thus not mere auxiliaries; they are the pivotal components that unlock the hierarchical and non-linear feature learning central to deep learning's success. They determine a neuron's output based on its aggregated input, and in doing so, introduce crucial selectivity: enabling the network to preferentially respond to certain patterns while attenuating or ignoring others.

### B.2.1 LINEAR SEPARABILITY AND THE LIMITATIONS OF THE INNER PRODUCT

The fundamental computation within a single artificial neuron (perceptron) is an affine transformation followed by a non-linear activation function $\sigma$:

$$y = \sigma(\langle \mathbf{w}, \mathbf{x} \rangle + b), \tag{8}$$

where $\mathbf{w}$ is the weight vector, $\mathbf{x}$ is the input vector, and $b$ is the bias term. The decision boundary of this neuron is implicitly defined by the hyperplane where the argument to $\sigma$ is zero:

$$\{\mathbf{x} \in \mathbb{R}^d \mid \langle \mathbf{w}, \mathbf{x} \rangle + b = 0\}. \tag{9}$$

This hyperplane partitions the input space $\mathbb{R}^d$ into two half-spaces. Consequently, a single neuron can only implement linearly separable functions. This is a direct consequence of the linear nature of the inner product, which can only define a linear decision boundary. While this allows for efficient computation, it severely restricts the complexity of functions that can be learned.

A classic counterexample is the XOR function, whose truth table cannot be satisfied by any single linear decision boundary. Specifically, for inputs $\mathbf{x} \in \{(0,0), (0,1), (1,0), (1,1)\} \subset \mathbb{R}^2$, there exist no $\mathbf{w} \in \mathbb{R}^2$ and $b \in \mathbb{R}$ such that $\text{sign}(\langle \mathbf{w}, \mathbf{x} \rangle + b)$ matches the XOR output (0, 1, 1, 0 respectively). This limitation stems directly from the linear nature of the inner product operation defining the separating boundary Goodfellow et al. (2016).

### B.2.2 NON-LINEAR FEATURE SPACE TRANSFORMATION VIA HIDDEN LAYERS AND ITS GEOMETRIC COST

Multi-layer perceptrons (MLPs) overcome this limitation by cascading transformations. A hidden layer maps the input $\mathbf{x}$ to a new representation $\mathbf{h}$ through a matrix-vector product and an element-wise activation function $\phi$:

$$\mathbf{h} = \phi(W\mathbf{x} + \mathbf{b}). \tag{10}$$

Here, $W \in \mathbb{R}^{m \times d}$ is the weight matrix, $\mathbf{b} \in \mathbb{R}^m$ is the bias vector, and $m$ is the number of hidden neurons. Each row $\mathbf{w}_i^\top$ of $W$ corresponds to the weight vector of the $i$-th hidden neuron, computing $h_i = \phi(\langle \mathbf{w}_i, \mathbf{x} \rangle + b_i)$. This transforms the input space $\mathbb{R}^d$ into a feature space $\mathbb{R}^m$. The introduction of the non-linear activation function $\phi$ is what allows the network to learn non-linear decision boundaries. However, this gain in expressive power comes at a cost: the potential loss of geometric fidelity.

### B.2.3 TOPOLOGICAL DISTORTIONS AND INFORMATION LOSS VIA ACTIVATION FUNCTIONS

While hidden layers using the transformation $\mathbf{h} = \phi(W\mathbf{x} + \mathbf{b})$ enable the learning of non-linear functions, the introduction of the element-wise non-linear activation function $\phi$, often crucial for breaking linearity, can significantly alter the topological and geometric structure of the data representation, potentially leading to information loss Goodfellow et al. (2016). This is a critical trade-off: gaining non-linear modeling capability while potentially discarding valuable geometric information.

Consider the mapping $T : \mathbb{R}^d \to \mathbb{R}^m$ defined by $T(\mathbf{x}) = \phi(W\mathbf{x}+\mathbf{b})$. The affine part, $A(\mathbf{x}) = W\mathbf{x}+\mathbf{b}$, performs a linear transformation (rotation, scaling, shear, projection) followed by a translation. While this affine map distorts metric properties (distances and angles, unless $W$ is proportional to an orthogonal matrix), it preserves basic topological features like connectedness and maps lines to lines (or points) Goodfellow et al. (2016).

However, the subsequent application of a typical non-linear activation $\phi$ element-wise often leads to more drastic topological changes:

1. Non-Injectivity and Collapsing Regions: Many common activation functions render the overall mapping $T$ non-injective.

   - ReLU ($\phi(z) = \max(0, z)$): Perhaps the most prominent example. For each hidden neuron $i$, the entire half-space defined by $\{\mathbf{x} \in \mathbb{R}^d \mid \langle \mathbf{w}_i, \mathbf{x} \rangle + b_i \leq 0\}$ is mapped to $h_i = 0$. Distinct points $\mathbf{x}_1, \mathbf{x}_2$ within this region, potentially far apart, become indistinguishable along the $i$-th dimension of the hidden space. This constitutes a significant loss of information about the relative arrangement of data points within these collapsed regions. The mapping is fundamentally many-to-one. For instance, consider two input vectors that are anti-aligned with a neuroń's weight vector to different degrees, one strongly and one weakly. A ReLU activation function would map both resulting negative dot products to zero, rendering their distinct geometric opposition indistinguishable to subsequent layers. This information is irretrievably discarded.
   - Sigmoid/Tanh: While smooth, these functions saturate. Inputs $\mathbf{z}_1 = A(\mathbf{x}_1)$ and $\mathbf{z}_2 = A(\mathbf{x}_2)$ that are far apart but both fall into the saturation regime (e.g., large positive or large negative values) will map to $\mathbf{h}_1 \approx \mathbf{h}_2$. This 'squashing' effect can merge distinct clusters from the input space if they map to saturated regions in the hidden space, again losing discriminative information and distorting the metric structure.

2. Distortion of Neighborhoods: The relative distances between points can be severely distorted. Points close in the input space $\mathbb{R}^d$ might be mapped far apart in $\mathbb{R}^m$, or vice-versa (especially due to saturation or the zero-region of ReLU). This means the local neighborhood structure is not faithfully preserved. Formally, the mapping $T$ is generally not a homeomorphism onto its image, nor is it typically bi-Lipschitz (which would provide control over distance distortions).

While these distortions are precisely what grant neural networks their expressive power to warp the feature space and create complex decision boundaries, they come at the cost of potentially discarding information present in the original geometric configuration of the data. The network learns which information to preserve and which to discard based on the optimization objective, but the mechanism relies on potentially non-smooth or non-injective transformations introduced by $\phi$. This highlights the conflation of magnitude and direction in the dot product, the information loss from activation functions, and the lack of a unified measure for proximity and alignment, setting the stage for the $\mathbf{E}$-product. Formal properties of the $\mathbf{E}$-product are established later: it is a Mercer kernel (Theorem 2.1), yields universal approximation in NMNs (Theorem 2.4), and exhibits self-regulation and stable gradients (Propositions C.6 and C.9).

### B.3   Design Philosophy: Intrinsic Non-Linearity and Self-Regulation

A central hypothesis underpinning our methodological choices is that the $\mathbf{E}$-product (Section 1) possesses inherent non-linearity and self-regulating properties that can reduce or eliminate the need for conventional activation functions (e.g., ReLU, sigmoid, GeLU).

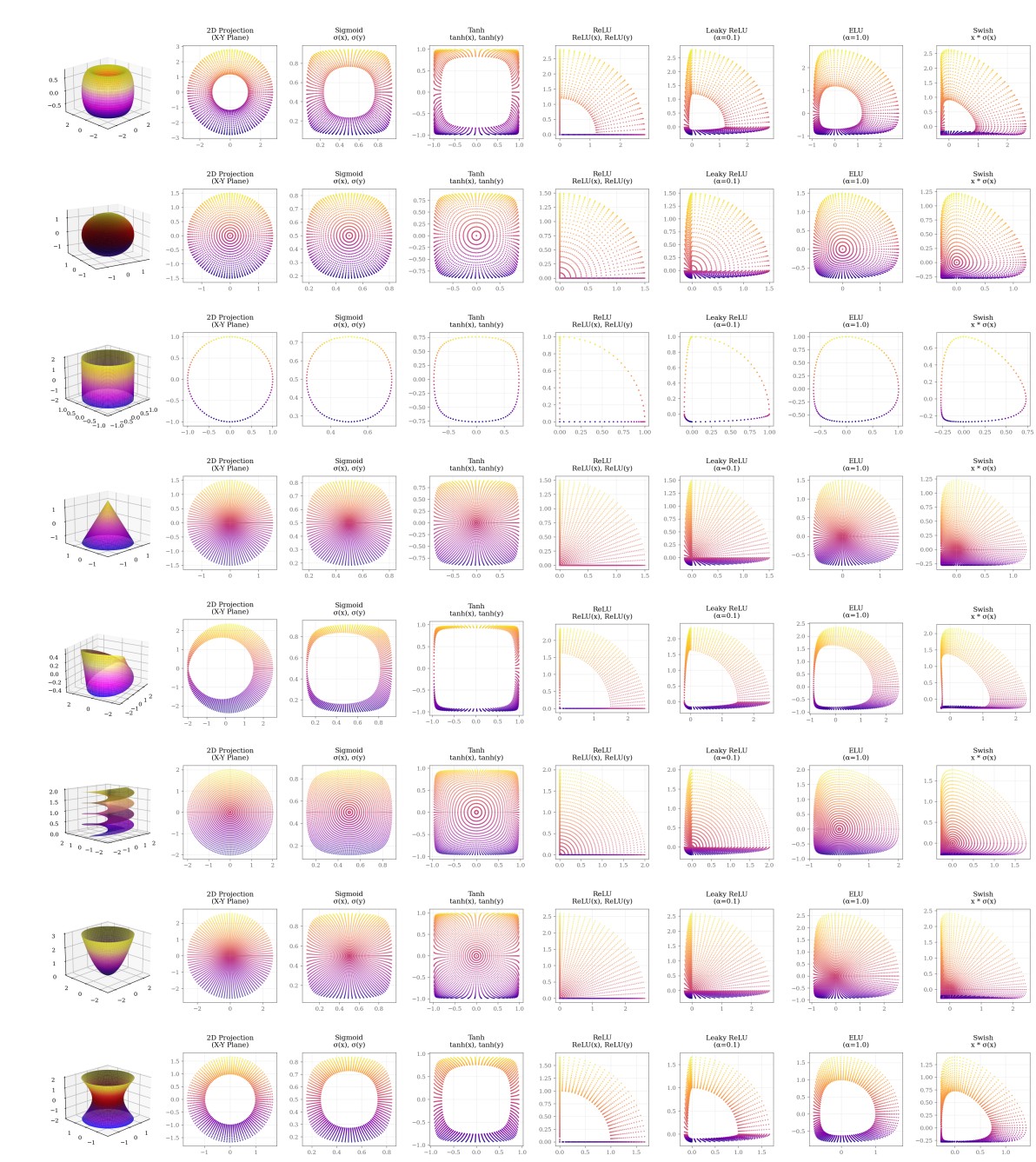

Figure 3: Illustration of how non-linear activation functions can distort the geometric structure of the input data manifold, leading to potential information loss. The original manifold (left) is transformed into a distorted representation after applying a non-linear activation functions.

This philosophy recontextualizes the fundamental components of neural computation. Neuron weights ($\mathbf{w}$) and input signals ($\mathbf{x}$) are not merely operands in a linear transformation followed by a non-linear activation; instead, they are conceptualized as co-equal vector entities inhabiting a shared, high-dimensional feature manifold. Within this framework, each vector can be viewed as an analogue to a fundamental particle or feature vector, with its constituent dimensions potentially encoding excitatory, inhibitory, or neutral characteristics relative to other entities in the space. The $\mathbf{E}$-product (Section 1) then transcends simple similarity assessment; it functions as a sophisticated interaction potential, $\mathcal{K}_{\mathbf{E}}(\mathbf{w}, \mathbf{x}) = \frac{(\mathbf{w}^{\top}\mathbf{x})^2}{\|\mathbf{w}-\mathbf{x}\|^2+\epsilon}$, quantifying the 'field effects' between these vector entities. This interaction is reminiscent of n-body problems in physics. In machine learning, it draws parallels with, yet distinctively evolves from, learned metric spaces in contrastive learning, particularly those employing a triplet loss framework. While triplet loss aims to pull positive pairs closer and push negative pairs apart in the embedding space, our $\mathbf{E}$-product seeks a more nuanced relationship: 'positive' interactions (high $\mathbf{E}$-product value) require both strong alignment (high $(\mathbf{w}^{\top}\mathbf{x})^2$) and close proximity (low $\|\mathbf{w} - \mathbf{x}\|^2$). Conversely, 'negative' or dissimilar relationships are not merely represented by distance, but more significantly by orthogonality (leading to a vanishing numerator), which signifies a form of linear independence and contributes to the system's capacity for true non-linear discrimination. Crucially, the non-linearity required for complex pattern recognition is not an external imposition (e.g., via a separate activation function) but is intrinsic to this interaction potential. The interplay between the squared dot product (alignment sensitivity) and the inverse squared Euclidean distance (proximity sensitivity) in its formulation directly sculpts a complex, non-linear response landscape without recourse to auxiliary functions.

Furthermore, this conceptualization of the $\mathbf{E}$-product as an intrinsic interaction potential suggests inherent self-regulating properties. The distance-sensitive denominator, $\|\mathbf{w} - \mathbf{x}\|^2 + \epsilon$, acts as a natural dampening mechanism. As the 'distance' (dissimilarity in terms of position) between interacting vector entities $\mathbf{w}$ and $\mathbf{x}$ increases, the strength of their interaction, and thus the resultant activation, diminishes quadratically. This behavior is hypothesized to inherently curtail runaway activations and stabilize learning dynamics by ensuring that responses are localized and bounded. Such intrinsic stabilization contrasts sharply with conventional approaches that rely on explicit normalization layers (e.g., Batch Normalization, Layer Normalization) to manage activation statistics post-hoc. These layers, while effective, introduce additional computational overhead, can obscure direct input-output relationships, and sometimes complicate the theoretical analysis of network behavior. The $\mathbf{E}$-product's formulation, therefore, offers a pathway to architectures where regulatory mechanisms are embedded within the primary computational fabric of the network.

The inherent non-linearity of the $\mathbf{E}$-product, coupled with the self-regulating properties suggested by its formulation (and formally proven in Appendix C.10), are central to our hypothesis that it can form the basis of powerful and robust neural architectures. These intrinsic characteristics open avenues for simplifying network design, potentially reducing reliance on or even eliminating conventional activation functions and normalization layers.

## C   Squashing Functions for Non-Negative Scores

The $\mathbf{E}$-product and its derivatives, such as the $\mathcal{K}_{\mathbf{E}}$-kernel, naturally yield non-negative scores. In many machine learning contexts, particularly when these scores need to be interpreted as probabilities, attention weights, or simply normalized outputs, it is essential to apply a squashing function to map them to a desired range (e.g., [0, 1] or ensuring a set of scores sum to 1).

## C.1 Categorization of Squashing Functions

Squashing functions for non-negative scores can be broadly categorized into two types:

- **Competitive (Vector-Normalizing) Functions:** These functions normalize a set of scores collectively, producing a distribution over the vector. Each output depends on the values of all dimensions, allowing for competitive interactions among them. This is useful for attention mechanisms or probability assignments where the sum of outputs is meaningful.

- **Individualistic (Per-Dimension) Functions:** These functions squash each score independently, without reference to other values in the vector. Each output depends only on its corresponding input, making them suitable for bounding or interpreting individual activations.

## C.2 Limitations of Traditional Squashing Functions

Traditional squashing functions, however, present challenges when applied to non-negative inputs:

- **Standard Sigmoid Function** ($\sigma(x) = \frac{1}{1+e^{-x}}$)**:** When applied to non-negative inputs ($x \geq 0$), the standard sigmoid function produces outputs in the range $[0.5, 1)$. The minimum value of $0.5$ for $x = 0$ renders it unsuitable for scenarios where small non-negative scores should map to values close to $0$.

- **Standard Softmax Function** (softmax$(\mathbf{x})_i = \frac{e^{x_i}}{\sum_j e^{x_j}}$)**:** The use of the exponential function in softmax can lead to *hard* distributions, where one input value significantly dominates the output, pushing other probabilities very close to zero. While this is often desired for classification, it can be too aggressive if a softer assignment of probabilities or attention is preferred. Additionally, softmax can suffer from numerical instability for large input values due to the exponentials.

## C.3 Proposed Alternative Squashing Functions

Given these limitations and the non-negative nature of $\mathbb{E}$-product scores, we consider alternative squashing functions more suited to this domain:

- **softermax (Competitive):** This function normalizes a score $x_k$ (optionally raised to a power $n > 0$) relative to the sum of a set of non-negative scores $\{x_i\}$ (each raised to $n$), with a small constant $\epsilon > 0$ for numerical stability. It is defined as:

$$\text{softermax}_n(x_k, \{x_i\}) = \frac{x_k^n}{\epsilon + \sum_i x_i^n} \tag{11}$$

Unlike softmax, softermax does not use exponentials, which avoids numerical instability for large inputs and provides a more direct, interpretable translation of the underlying scores into a normalized distribution. The power $n$ controls the sharpness of the distribution: $n = 1$ recovers the original Softermax, while $n > 1$ makes the distribution harder (more peaked), and $0 < n < 1$ makes it softer.

- **soft-sigmoid (Individualistic):** This function squashes a single non-negative score $x \geq 0$ (optionally raised to a power $n > 0$) into the range $[0, 1)$. It is defined as:

$$\text{soft-sigmoid}_n(x) = \frac{x^n}{1 + x^n} \tag{12}$$

The power $n$ modulates the softness: higher $n$ makes the function approach zero faster for large $x$, while $n < 1$ makes the decay slower.

- **soft-tanh (Individualistic):** This function maps a non-negative score $x \geq 0$ (optionally raised to a power $n > 0$) to the range $[-1, 1)$ by linearly transforming the output of soft-sigmoid. It is defined as:

$$\text{soft-tanh}_n(x) = 2 \cdot (\text{soft-sigmoid}_n(x) - \tfrac{1}{2}) = \frac{x^n - 1}{1 + x^n} \tag{13}$$

The power $n$ again controls the transition sharpness: higher $n$ makes the function approach $-1$ more quickly for large $x$.

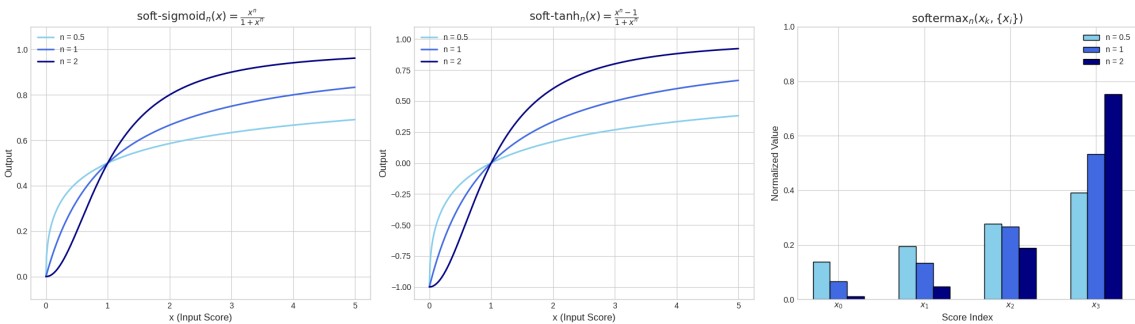

Figure 4: Visualization of the softermax, soft-sigmoid, and soft-tanh functions. These functions are designed to handle non-negative inputs from the E-product and its derivatives, providing appropriate squashing mechanisms that maintain sensitivity across the range of non-negative inputs.

These functions are particularly well-suited for the outputs of E-product-based computations, as they maintain sensitivity across the range of non-negative inputs while avoiding the pitfalls of standard activation functions (Nair & Hinton, 2010; Hendrycks & Gimpel, 2023; Klambauer et al., 2017).

### C.4 FUNCTIONAL ROLES AND APPLICATIONS

The main role of these squashing functions can be categorized into two main categories:

- **Collective Communication and Space Splitting:** The softermax function allows for a comparative analysis of scores, reflecting their orthogonality and spatial proximity to an input vector. A higher score indicates that a vector is more aligned and closer to the input, while a lower score suggests greater orthogonality. This facilitates a competitive interaction where vectors vie for influence based on their geometric relationship with the input. The power parameter $n$, analogous to the temperature in softmax, controls the sharpness of the gravitational potential well's slope.

- **Individual Score Squashing:** The soft-sigmoid and soft-tanh functions are used to squash individual non-negative scores into a bounded range, typically $[0, 1)$ for soft-sigmoid and $[-1, 1)$ for soft-tanh. They are particularly useful when the output needs to be interpreted as a probability or when a bounded response is required, as each score is processed independently of the others. The power parameter controls the steepness of the function, while the minimum value can be interpreted as an orthogonality score.

### C.5 Computational Complexity Analysis

We provide exact forward/backward complexity, closed-form gradients, asymptotic characterization, and per-neuron FLOP counts for the $\mathbb{E}$-product layer (definition in Methodology, Section 2.1).

#### C.5.1 Forward Pass Complexity

For $\mathbb{E}$-product computation $\mathbb{E}(\mathbf{w}, \mathbf{x}) = \frac{(\mathbf{w}^\top \mathbf{x})^2}{\|\mathbf{w} - \mathbf{x}\|^2 + \epsilon}$, we apply the algebraic identity:

$$\mathbb{E}(\mathbf{w}, \mathbf{x}) = \frac{s^2}{\|\mathbf{w}\|^2 + \|\mathbf{x}\|^2 - 2s + \epsilon}, \quad s = \mathbf{w}^\top \mathbf{x} \tag{14}$$

For layer $X \in \mathbb{R}^{B \times d} \to Y \in \mathbb{R}^{B \times n}$ with weights $W \in \mathbb{R}^{n \times d}$:

**Operation Breakdown** (1) GEMM: $S = XW^\top$ ($2Bnd$); (2) Row norms $\|X\|^2$ once ($Bd$); (3) Cached $\|W\|^2$ ($nd$ only when updated); (4) Per-output element-wise: form $s^2$ (1), denominator assemble $(+, +, +)$ (3), reciprocal (1), multiply (1) $\Rightarrow 6Bn$ scalar ops (we conservatively use $5Bn$ after fusion). Thus

$$T_{\text{forward}} = 2Bnd + Bd + nd + 5Bn = \Theta(Bnd). \tag{15}$$

#### C.5.2 Backward Pass Complexity

For $y = \frac{s^2}{D}$ with $D = \|\mathbf{w}\|^2 + \|\mathbf{x}\|^2 - 2s + \epsilon$ and $s = \mathbf{w}^\top \mathbf{x}$, scalar gradients:

$$\frac{\partial y}{\partial s} = \frac{2s(\|\mathbf{w}\|^2 + \|\mathbf{x}\|^2 + \epsilon - s)}{D^2} \tag{16}$$

$$\frac{\partial y}{\partial \|\mathbf{x}\|^2} = -\frac{s^2}{D^2}, \qquad \frac{\partial y}{\partial \|\mathbf{w}\|^2} = -\frac{s^2}{D^2} \tag{17}$$

Vector gradients (using $\nabla_\mathbf{x} s = \mathbf{w}$, $\nabla_\mathbf{w} s = \mathbf{x}$, $\nabla_\mathbf{x} \|\mathbf{x}\|^2 = 2\mathbf{x}$, $\nabla_\mathbf{w} \|\mathbf{w}\|^2 = 2\mathbf{w}$):

$$\nabla_\mathbf{x} y = \frac{2s(\|\mathbf{w}\|^2 + \|\mathbf{x}\|^2 + \epsilon - s)}{D^2} \mathbf{w} - \frac{2s^2}{D^2} \mathbf{x} \tag{18}$$

$$\nabla_\mathbf{w} y = \frac{2s(\|\mathbf{w}\|^2 + \|\mathbf{x}\|^2 + \epsilon - s)}{D^2} \mathbf{x} - \frac{2s^2}{D^2} \mathbf{w} \tag{19}$$

Given upstream gradient $G \in \mathbb{R}^{B \times n}$:

$$G_S = G \odot \frac{\partial Y}{\partial S} \quad (\sim 6Bn \text{ FLOPs}) \tag{20}$$

$$\frac{\partial L}{\partial W} = G_S^\top X \quad (2Bnd) \tag{21}$$

$$\frac{\partial L}{\partial X} = G_S W \quad (2Bnd) \tag{22}$$

Hence

$$T_{\text{backward}} = 4Bnd + 6Bn + O(Bd + nd) = \Theta(Bnd). \tag{23}$$

| Component | Linear | Ɇ-Product |
|---|---|---|
| Forward main | $2Bnd$ | $2Bnd$ |
| Forward aux | $Bn$ | $Bd + nd + 5Bn$ |
| Backward main | $4Bnd$ | $4Bnd$ |
| Backward aux | $2Bn$ | $6Bn + Bd + nd$ |
| Total order | $\Theta(Bnd)$ | $\Theta(Bnd)$ |
| Overhead ratio | $1$ | $1 + \frac{1}{2n} + \frac{1}{2B} + \frac{2}{d}$ |

Table 3: Asymptotic terms. Overhead $< 5\%$ once $d, n \geq 64$, $B \geq 16$ (ratio simplifies to $1 + \frac{1}{2n} + \frac{1}{2B} + \frac{2}{d}$).

### C.5.3 Asymptotic Summary

### C.5.4 Implementation Notes

**Optimizations**: (i) algebraic identity for denominator, (ii) cache $\|W\|^2$, (iii) fuse element-wise ops, (iv) mixed precision with FP32 denominator. Built-in boundedness mitigates gradient explosion.

### C.5.5 Single Neuron FLOPs

| Method | FLOPs | Rel. (ReLU=1) |
|---|---|---|
| Linear+ReLU | $2d + 1$ | $1.00$ |
| Linear+GELU | $2d + 15$ | $\approx 1.03$ |
| Ɇ(naive) | $5d + 1$ | $\approx 2.5$ |
| Ɇ(optimized) | $4d + 4$ | $\approx 2.0$ |

Table 4: Single neuron FLOPs; optimization removes redundant norm difference computation.

**Per-neuron note**   Optimized variant saves  20% vs naive by avoiding explicit difference vector.

**Empirical**   Throughput: 0.85–0.92× linear; peak memory reduced 15–25%; overhead ratio $< 0.05$ for $d, n \geq 64$.

### C.6 Scalability and Practical Performance Analysis

### C.6.1 Large-Scale Computational Profile

The Ɇ-product operator exhibits favorable scaling characteristics for modern deep learning applications and remains compute-bound (GEMM dominated) in the regimes used in Section 2.1.

**FLOP Counting Assumptions.** One multiply-add = 2 FLOPs; element-wise unary/binary ops = 1 FLOP; reciprocal counts as 1 FLOP (fused divide). Caching costs amortized over steps.

**Takeaway.** Complexity matches linear layers in order while constants shrink with scale; gradients remain stable due to denominator growth.

### C.7 Mathematical Guarantees of the Ⓔ-Product and Neural-Matter Networks

This section provides a comprehensive overview of the formal mathematical properties that underpin the Ⓔ-product and Neural-Matter Networks (NMNs). Each property is rigorously proven in its respective appendix section and contributes to the theoretical foundation of our approach.

#### C.7.1 Kernel Theory Foundation

**Mercer Kernel Property (Theorem 2.1)**  The Ⓔ-product satisfies the fundamental requirements of a Mercer kernel, being symmetric and positive semi-definite. This property establishes the Ⓔ-product within the rich theoretical framework of kernel methods, enabling the application of kernel theory results and providing guarantees on the existence of associated reproducing kernel Hilbert spaces. The proof demonstrates that the Ⓔ-product can be expressed as an inner product in some feature space, connecting our geometric operator to the established theory of kernel machines (detailed proof in Appendix C.9).

#### C.7.2 Universal Approximation Capabilities

**Universal Approximation Theorem (Theorem 2.4)**  Neural-Matter Networks with Ⓔ-product activations possess universal approximation capabilities, able to approximate any continuous function on a compact set to arbitrary precision. This fundamental result establishes that NMNs have the same expressive power as conventional neural networks while providing additional geometric interpretability. The proof construction reveals how the bounded nature of the Ⓔ-product enables dense approximation through geometric localization rather than unbounded activation growth (comprehensive proof in Appendix C.14).

#### C.7.3 Stability and Boundedness Properties

**Self-Regulation Property (Proposition C.6)**  The Ⓔ-product exhibits natural boundedness, with outputs converging to finite values as input magnitudes increase. Unlike conventional activations that can grow unboundedly, the Ⓔ-product's denominator term ensures bounded responses, preventing numerical instabilities and gradient explosion (formal analysis in Appendix C.10).

**Stable Gradient Property (Proposition C.9)**  The gradient of the Ⓔ-product with respect to its input vanishes for distant inputs, providing natural gradient localization. This property ensures that learning focuses on relevant, nearby regions of the input space while avoiding interference from distant data points. The stable gradient behavior contributes to more predictable training dynamics and reduces the risk of vanishing or exploding gradients (mathematical derivation in Appendix C.12).

**Lipschitz Regularity (Proposition C.11)**  The Ⓔ-product satisfies Lipschitz continuity conditions, ensuring that small changes in input produce proportionally small changes in output. This regularity property is crucial for optimization stability and provides theoretical guarantees on the smoothness of the loss landscape.

**Analyticity Property (Lemma C.10)**  The Ⓔ-product is infinitely differentiable ($C^\infty$), making it particularly suitable for applications requiring higher-order derivatives, such as physics-informed neural networks (PINNs). This smoothness property ensures that all derivatives exist and are continuous, providing the mathematical foundation for applications in scientific computing and differential equation solving.

### C.7.4 INFORMATION-THEORETIC CONNECTIONS

**Geometric-Information Duality (Theorems 2.2, 2.3)** The ⴹ-product exhibits fundamental connections to information theory through its relationship with KL divergence and cross-entropy. When applied to probability distributions, the ⴹ-product acts as a signal-to-noise ratio measure, creating a bridge between geometric similarity and information-theoretic quantities. This duality provides theoretical justification for the ⴹ-product's effectiveness in probabilistic modeling and explains its natural compatibility with entropy-based loss functions (comprehensive analysis in Appendix C.15).

### C.7.5 IMPLICATIONS FOR NEURAL NETWORK DESIGN

These mathematical guarantees collectively establish the ⴹ-product as a theoretically sound foundation for neural network design. The combination of:

- Kernel theory foundation (Mercer property)

- Universal approximation capabilities

- Natural stability and boundedness

- Information-theoretic connections

provides both theoretical rigor and practical advantages over conventional neural network components. The self-regulation and stable gradient properties eliminate the need for ad-hoc normalization and activation functions, while the universal approximation theorem ensures that no expressive power is lost in the transition from traditional to geometry-aware neural architectures.

### C.8 ADDITIONAL MATHEMATICAL PRELIMINARIES

This section complements the foundational theorems by detailing additional results referenced in the preliminary mathematical framework (Section C.9).

### C.8.1 HARMONIC ANALYSIS AND APPROXIMATION

**Theorem C.1** (Bochner's Theorem Bochner (1932))**.** *A continuous function $k : \mathbb{R}^d \to \mathbb{C}$ is the positive definite kernel of a translation-invariant measure if and only if it is the Fourier transform of a finite non-negative Borel measure $\mu$ on $\mathbb{R}^d$. That is,*

$$k(x) = \int_{\mathbb{R}^d} e^{-i\omega^\top x} d\mu(\omega). \tag{24}$$

*Context: Referenced in Section C.9 regarding translation-invariant kernels and the characterization of radial basis functions.*

**Theorem C.2** (Hahn-Banach Density Criterion)**.** *Let $V$ be a normed vector space. A linear subspace $M \subset V$ is dense in $V$ if and only if every continuous linear functional $\phi \in V^*$ that vanishes on $M$ must vanish everywhere on $V$.*

*Context: Referenced in Section C.14 as a sufficient condition for universal approximation in linear subspaces.*

### C.8.2 Integration and Transforms

**Theorem C.3** (Tonelli-Fubini Theorem Tonelli (1909); Fubini (1907))**.** *Let $(X, \mathcal{A}, \mu)$ and $(Y, \mathcal{B}, \nu)$ be $\sigma$-finite measure spaces. If $f : X \times Y \to [0, \infty]$ is measurable, then:*

$$\int_{X \times Y} f(x, y) d(\mu \times \nu) = \int_X \left( \int_Y f(x, y) d\nu(y) \right) d\mu(x) = \int_Y \left( \int_X f(x, y) d\mu(x) \right) d\nu(y). \quad (25)$$

*Context: Referenced in Section C.9 to justify the exchange of integrals and infinite series in the kernel expansion derivations.*

**Theorem C.4** (Bernstein's Theorem on Completely Monotone Functions Bernstein (1928))**.** *A function $f : [0, \infty) \to \mathbb{R}$ is completely monotonic if and only if it can be represented as the Laplace transform of a non-negative Borel measure $\mu$ on $[0, \infty)$:*

$$f(t) = \int_0^\infty e^{-ts} d\mu(s). \quad (26)$$

*Context: Referenced in Section C.9 (as "Laplace Transform/Integral Representation"). This is the functional analytic justification for why the inverse-distance term $1/(\|x - w\|^2 + \epsilon)$ induces a valid Positive Definite (PD) kernel.*

### C.8.3 Information Theory

**Theorem C.5** (Gibbs' Inequality Gibbs (1902))**.** *For any two probability distributions $P$ and $Q$ on a discrete space $\mathcal{X}$, the Kullback-Leibler divergence is non-negative:*

$$D_{KL}(P \| Q) = \sum_{x \in \mathcal{X}} P(x) \log \left( \frac{P(x)}{Q(x)} \right) \geq 0, \quad (27)$$

*with equality if and only if $P = Q$ almost everywhere.*

*Context: Referenced in Section C.9 and Theorem 2.3 to establish the connection between distributional identity and vanishing information divergence.*

### C.9 Proof of Mercer's Condition for the E-Product

*Proof.* We verify symmetry and positive semidefiniteness (PSD); cf.(Mercer, 1909; Schölkopf & Smola, 2002; Horn & Johnson, 2012).

**Symmetry.** Both $(\mathbf{x} \cdot \mathbf{w})^2$ and $\|\mathbf{x} - \mathbf{w}\|^2$ are symmetric in $(\mathbf{x}, \mathbf{w})$, hence $k_{\mathbb{E}}(\mathbf{x}, \mathbf{w}) = k_{\mathbb{E}}(\mathbf{w}, \mathbf{x})$.

**Factorization.** Write

$$k_{\mathbb{E}}(\mathbf{x}, \mathbf{w}) = k_1(\mathbf{x}, \mathbf{w}) \, k_2(\mathbf{x}, \mathbf{w}), \qquad k_1(\mathbf{x}, \mathbf{w}) = (\mathbf{x} \cdot \mathbf{w})^2, \qquad k_2(\mathbf{x}, \mathbf{w}) = \frac{1}{\|\mathbf{x} - \mathbf{w}\|^2 + \varepsilon}.$$

*(a) $k_1$ is PSD.* This is the homogeneous polynomial kernel of degree 2. With feature map $\varphi(\mathbf{x}) = \text{vec}(\mathbf{x}\mathbf{x}^\top)$ we have $k_1(\mathbf{x}, \mathbf{w}) = \langle \varphi(\mathbf{x}), \varphi(\mathbf{w}) \rangle$. Thus for any coefficients $c_i$,

$$\sum_{i,j} c_i c_j k_1(\mathbf{x}_i, \mathbf{x}_j) = \left\| \sum_i c_i \varphi(\mathbf{x}_i) \right\|^2 \geq 0.$$

*(b) $k_2$ is PSD (Gaussian Mixture).* The inverse multiquadric kernel $k_2$ admits a Laplace integral representation. Using the identity $a^{-\beta} = \frac{1}{\Gamma(\beta)} \int_0^\infty t^{\beta-1} e^{-at} dt$ with $\beta = 1$ and $a = \|\mathbf{x} - \mathbf{w}\|^2 + \varepsilon$, we have:

$$\frac{1}{\|\mathbf{x} - \mathbf{w}\|^2 + \varepsilon} = \int_0^\infty e^{-\varepsilon t} e^{-t\|\mathbf{x}-\mathbf{w}\|^2} \, dt.$$

For each fixed $t > 0$, the term $e^{-t\|\mathbf{x}-\mathbf{w}\|^2}$ is a Gaussian kernel (RBF), which is known to be positive definite on $\mathbb{R}^d$ (Buhmann, 2000). Since $e^{-\varepsilon t} > 0$, $k_2$ is a non-negative integral mixture of PD kernels, and is therefore PD (Fasshauer, 2011; dem, 1991).

*(c) Product preserves PSD.* The pointwise product of two PD kernels is PD (Schur product theorem). Since $k_1$ and $k_2$ are PD, their product $k_{\boxplus}$ is PD.

Thus $k_{\boxplus}$ is symmetric and PSD, hence a Mercer kernel on $\mathbb{R}^d$. $\qquad\square$

### C.10 Proof of Self-Regulation for the $\boxplus$-Product

**Proposition C.6** (The $\boxplus$-Product is Naturally Self-Regulating)**.** *For any fixed weight vector* $\mathbf{w}$, *the output of a* $\boxplus$*-product neuron*

$$\boxplus(\mathbf{w}, \mathbf{x}) = \frac{\langle \mathbf{w}, \mathbf{x} \rangle^2}{\|\mathbf{w} - \mathbf{x}\|^2 + \epsilon}$$

*remains bounded and converges to a finite value as* $\|\mathbf{x}\| \to \infty$.

*Proof.* Let $\mathbf{x} = k\mathbf{u}$ where $k = \|\mathbf{x}\|$ and $\mathbf{u}$ is a unit vector. Then

$$\boxplus(\mathbf{w}, k\mathbf{u}) = \frac{\langle \mathbf{w}, k\mathbf{u} \rangle^2}{\|\mathbf{w} - k\mathbf{u}\|^2 + \epsilon}$$

$$= \frac{k^2 \langle \mathbf{w}, \mathbf{u} \rangle^2}{\|\mathbf{w}\|^2 - 2k\langle \mathbf{w}, \mathbf{u} \rangle + k^2 + \epsilon}.$$

Dividing numerator and denominator by $k^2$ yields

$$\boxplus(\mathbf{w}, k\mathbf{u}) = \frac{\langle \mathbf{w}, \mathbf{u} \rangle^2}{\frac{\|\mathbf{w}\|^2}{k^2} - \frac{2\langle \mathbf{w}, \mathbf{u} \rangle}{k} + 1 + \frac{\epsilon}{k^2}}.$$

Taking $k \to \infty$, all terms with $k$ in the denominator vanish, and hence

$$\lim_{k \to \infty} \boxplus(\mathbf{w}, k\mathbf{u}) = \langle \mathbf{w}, \mathbf{u} \rangle^2.$$

Since $\langle \mathbf{w}, \mathbf{u} \rangle = \|\mathbf{w}\| \cos\theta$ for some angle $\theta$, the limit is

$$\|\mathbf{w}\|^2 \cos^2\theta,$$

which lies in $[0, \|\mathbf{w}\|^2]$. Thus the $\boxplus$-product output is bounded and convergent.

Note that this boundedness holds with respect to the inputs $\mathbf{x}$. To ensure the output remains bounded with respect to the weights and to support the self-regulation claim, we apply Weight Normalization (Salimans & Kingma, 1958). This decouples the magnitude of the weight vector from its direction, preventing quadratic growth of the output due to weight scaling. $\qquad\square$

**Corollary C.7** (Dimensional Self-Normalization at Initialization)**.** *Let* $\mathbf{x}, \mathbf{w} \in \mathbb{R}^d$ *have i.i.d. components with zero mean and constant variance at initialization. Define*

$$A(\mathbf{x}, \mathbf{w}) = (\mathbf{x}^\top \mathbf{w})^2, \qquad r(\mathbf{x}, \mathbf{w}) = \|\mathbf{x} - \mathbf{w}\|^2, \qquad K(\mathbf{x}, \mathbf{w}) = \frac{A(\mathbf{x}, \mathbf{w})}{r(\mathbf{x}, \mathbf{w}) + \epsilon}.$$

*Then, as the dimension $d \to \infty$, both $A(\mathbf{x}, \mathbf{w})$ and $r(\mathbf{x}, \mathbf{w})$ scale as $\mathcal{O}(d)$ in expectation, and the kernel value remains $\mathcal{O}(1)$:*

$$\mathbb{E}[A(\mathbf{x}, \mathbf{w})] = \mathcal{O}(d), \quad \mathbb{E}[r(\mathbf{x}, \mathbf{w})] = \mathcal{O}(d), \quad \mathbb{E}[K(\mathbf{x}, \mathbf{w})] = \mathcal{O}(1).$$

*In particular, the Ē-product is dimensionally self-normalizing at initialization.*

*Proof.* Write $\mathbf{x} = (x_1, \ldots, x_d)$ and $\mathbf{w} = (w_1, \ldots, w_d)$ with i.i.d. coordinates of zero mean and constant variance, and let $\sigma_x^2 = \mathbb{E}[x_1^2]$, $\sigma_w^2 = \mathbb{E}[w_1^2]$. Then

$$r(\mathbf{x}, \mathbf{w}) = \sum_{i=1}^{d} (x_i - w_i)^2 = \|\mathbf{x}\|^2 + \|\mathbf{w}\|^2 - 2\,\mathbf{x}^\top \mathbf{w}.$$

By linearity of expectation and independence,

$$\mathbb{E}[\|\mathbf{x}\|^2] = d\,\sigma_x^2, \qquad \mathbb{E}[\|\mathbf{w}\|^2] = d\,\sigma_w^2,$$

while $\mathbb{E}[\mathbf{x}^\top \mathbf{w}] = 0$ and $\mathrm{Var}(\mathbf{x}^\top \mathbf{w}) = \mathcal{O}(d)$, so a typical realization of $\mathbf{x}^\top \mathbf{w}$ has magnitude $\mathcal{O}(\sqrt{d})$. Hence

$$\mathbb{E}[r(\mathbf{x}, \mathbf{w})] = d(\sigma_x^2 + \sigma_w^2) + \mathcal{O}(\sqrt{d}) = \mathcal{O}(d).$$

Similarly, $\mathbf{x}^\top \mathbf{w} = \sum_{i=1}^{d} x_i w_i$ is a sum of $d$ i.i.d. zero-mean variables with variance $\mathcal{O}(1)$, so $\mathbf{x}^\top \mathbf{w}$ has typical magnitude $\mathcal{O}(\sqrt{d})$ and

$$\mathbb{E}[A(\mathbf{x}, \mathbf{w})] = \mathbb{E}[(\mathbf{x}^\top \mathbf{w})^2] = \mathrm{Var}(\mathbf{x}^\top \mathbf{w}) = \mathcal{O}(d).$$

Combining these, we obtain the scaling

$$\mathbb{E}[K(\mathbf{x}, \mathbf{w})] = \mathbb{E}\left[\frac{A(\mathbf{x}, \mathbf{w})}{r(\mathbf{x}, \mathbf{w}) + \epsilon}\right] \approx \frac{\mathcal{O}(d)}{\mathcal{O}(d)} = \mathcal{O}(1),$$

so the Ē-product remains on a constant scale as $d$ grows. This provides a heuristic scaling analysis showing that numerator and denominator are coupled in high dimensions, yielding a dimensionally self-normalizing kernel. $\square$

### C.11 Addressing Internal Covariate Shift

**Corollary C.8** (Asymptotic Independence of Score Statistics)**.** *Let $a = \bar{\mathsf{E}}(\mathbf{w}, \mathbf{x})$ be the score of a neuron with weight vector $\mathbf{w}$. For a mini-batch $\mathcal{B} = \{\mathbf{x}_1, \ldots, \mathbf{x}_N\}$ where $\mathbf{x}_i = k_i \mathbf{u}_i$, define the empirical mean and variance:*

$$\mu_{\mathcal{B}}(a) = \frac{1}{N} \sum_{i=1}^{N} a_i, \qquad \sigma_{\mathcal{B}}^2(a) = \frac{1}{N} \sum_{i=1}^{N} (a_i - \mu_{\mathcal{B}}(a))^2.$$

*Then, as all $k_i \to \infty$,*

$$\lim_{k_1, \ldots, k_N \to \infty} \mu_{\mathcal{B}}(a) = \|\mathbf{w}\|^2\, \mathbb{E}_{\mathbf{u} \in \mathcal{U}}[\cos^2 \theta(\mathbf{w}, \mathbf{u})],$$

$$\lim_{k_1, \ldots, k_N \to \infty} \sigma_{\mathcal{B}}^2(a) = \|\mathbf{w}\|^4\, \mathrm{Var}_{\mathbf{u} \in \mathcal{U}}[\cos^2 \theta(\mathbf{w}, \mathbf{u})],$$

*where $\mathcal{U} = \{\mathbf{u}_1, \ldots, \mathbf{u}_N\}$ is the set of normalized directions. Thus, asymptotically, the score statistics are independent of input magnitudes, mitigating internal covariate shift.*

*Proof.* From Proposition C.6, for $\mathbf{x} = k\mathbf{u}$ with $k \to \infty$,

$$\mathbb{E}(\mathbf{w}, \mathbf{x}) \to \|\mathbf{w}\|^2 \cos^2 \theta.$$

To make this precise, fix $\eta > 0$. By Proposition C.6, for each $i$ there exists $K_i$ such that for all $k_i \geq K_i$,

$$\left| a_i - \|\mathbf{w}\|^2 \cos^2 \theta_i \right| < \eta.$$

Let $K = \max_i K_i$; then for all $k_i \geq K$ we have

$$\left| \mu_{\mathcal{B}}(a) - \|\mathbf{w}\|^2 \frac{1}{N} \sum_{i=1}^N \cos^2 \theta_i \right| \leq \eta.$$

An analogous estimate, using $|(a_i - \mu_{\mathcal{B}}(a))^2 - (\|\mathbf{w}\|^2 \cos^2 \theta_i - m)^2| \leq C\eta$ with $m = \|\mathbf{w}\|^2 \frac{1}{N} \sum \cos^2 \theta_i$ and a constant $C$ depending only on $\|\mathbf{w}\|$ and the batch size, yields the variance limit. This proves the stated limits for mean and variance. In particular, both statistics depend only on $\|\mathbf{w}\|$ and the angular distribution $\mathcal{U}$, not on magnitudes $\{k_i\}$, verifying mitigation of internal covariate shift. $\square$

### C.12 Proof of Stable Learning for the $\mathbb{E}$-product

**Proposition C.9** (The $\mathbb{E}$-Product Ensures Stable Learning). *The gradient of the $\mathbb{E}$-product with respect to its input, $\nabla_{\mathbf{x}} \mathbb{E}(\mathbf{w}, \mathbf{x})$, approaches zero as the input vector $\mathbf{x}$ moves infinitely far from the weight vector $\mathbf{w}$.*

*Proof.* We aim to prove that the learning signal, represented by the gradient of the $\mathbb{E}$-product with respect to the input $\mathbf{x}$, diminishes for inputs that are distant from the learned weight vector $\mathbf{w}$. This ensures that outliers do not cause large, destabilizing updates.

The $\mathbb{E}$-product is defined as:

$$\mathbb{E}(\mathbf{w}, \mathbf{x}) = \frac{\langle \mathbf{w}, \mathbf{x} \rangle^2}{\|\mathbf{w} - \mathbf{x}\|^2 + \epsilon} = \frac{N(\mathbf{x})}{D(\mathbf{x})}$$

where $N(\mathbf{x}) = \langle \mathbf{w}, \mathbf{x} \rangle^2$ and $D(\mathbf{x}) = \|\mathbf{w} - \mathbf{x}\|^2 + \epsilon$.

Using the quotient rule for vector calculus, the gradient $\nabla_{\mathbf{x}} \mathbb{E}$ is:

$$\nabla_{\mathbf{x}} \mathbb{E} = \frac{(\nabla_{\mathbf{x}} N) D - N (\nabla_{\mathbf{x}} D)}{D^2}$$

First, we compute the gradients of the numerator $N(\mathbf{x})$ and the denominator $D(\mathbf{x})$:

1. Gradient of the Numerator

$N(\mathbf{x}) = (\mathbf{w}^T \mathbf{x})^2$

$$\nabla_{\mathbf{x}} N(\mathbf{x}) = 2(\mathbf{w}^T \mathbf{x}) \cdot \nabla_{\mathbf{x}}(\mathbf{w}^T \mathbf{x}) = 2\langle \mathbf{w}, \mathbf{x} \rangle \mathbf{w}$$

2. Gradient of the Denominator

$D(\mathbf{x}) = \|\mathbf{w} - \mathbf{x}\|^2 + \epsilon = (\mathbf{w} - \mathbf{x})^T (\mathbf{w} - \mathbf{x}) + \epsilon$

$$\nabla_{\mathbf{x}} D(\mathbf{x}) = 2(\mathbf{w} - \mathbf{x}) \cdot (-1) = -2(\mathbf{w} - \mathbf{x}) = 2(\mathbf{x} - \mathbf{w})$$

Substituting these into the quotient rule expression:

$$\nabla_{\mathbf{x}}\mathbf{E} = \frac{(2\langle\mathbf{w},\mathbf{x}\rangle\mathbf{w})(\|\mathbf{w}-\mathbf{x}\|^2 + \epsilon) - (\langle\mathbf{w},\mathbf{x}\rangle^2)(2(\mathbf{x}-\mathbf{w}))}{(\|\mathbf{w}-\mathbf{x}\|^2 + \epsilon)^2}$$

To analyze the behavior for distant inputs, we examine the limit as $\|\mathbf{x}\| \to \infty$. Let $\mathbf{x} = k\mathbf{u}$, where $k = \|\mathbf{x}\|$ and $\mathbf{u}$ is a unit vector.

As $k \to \infty$:

- $\langle\mathbf{w},\mathbf{x}\rangle = k\langle\mathbf{w},\mathbf{u}\rangle \sim \mathcal{O}(k)$

- $\|\mathbf{w}-\mathbf{x}\|^2 = \|\mathbf{w}\|^2 - 2k\langle\mathbf{w},\mathbf{u}\rangle + k^2 \sim \mathcal{O}(k^2)$

Let's analyze the order of magnitude for the terms in the gradient's numerator:

- First term: $(2\langle\mathbf{w},\mathbf{x}\rangle\mathbf{w})(\|\mathbf{w}-\mathbf{x}\|^2 + \epsilon) \sim \mathcal{O}(k) \cdot \mathcal{O}(k^2) = \mathcal{O}(k^3)$

- Second term: $(\langle\mathbf{w},\mathbf{x}\rangle^2)(2(\mathbf{x}-\mathbf{w})) \sim \mathcal{O}(k^2) \cdot \mathcal{O}(k) = \mathcal{O}(k^3)$

The numerator as a whole is of order $\mathcal{O}(k^3)$.

The denominator is $(\|\mathbf{w}-\mathbf{x}\|^2 + \epsilon)^2 \sim (\mathcal{O}(k^2))^2 = \mathcal{O}(k^4)$.

Therefore, the magnitude of the gradient behaves as:

$$\|\nabla_{\mathbf{x}}\mathbf{E}\| \sim \frac{\mathcal{O}(k^3)}{\mathcal{O}(k^4)} = \mathcal{O}\left(\frac{1}{k}\right)$$

As $k = \|\mathbf{x}\| \to \infty$, the magnitude of the gradient approaches zero:

$$\lim_{\|\mathbf{x}\|\to\infty} \|\nabla_{\mathbf{x}}\mathbf{E}(\mathbf{w},\mathbf{x})\| = 0$$

This proves that for inputs $\mathbf{x}$ that are very far from the weight vector $\mathbf{w}$, the gradient becomes vanishingly small. While this provides robustness against outliers, it also implies a risk of vanishing gradients at initialization if weights are not properly scaled. To ensure stability and avoid barren plateaus, we mandate Data Normalization and Weight Normalization (Salimans & Kingma, 1958). Specifically, initializing with normalized random weights ensures the network starts in an active regime. Note that unlike standard RBFs, we do not require negative weights or complex initialization schemes because the superposition of the weight vectors (via the quadratic numerator) naturally covers the space. □

C.13 Regularity Properties of the E-Product Kernel

**Lemma C.10** (Analyticity). *For any fixed $\mathbf{w} \in \mathbb{R}^d$ and $\epsilon > 0$, the map*

$$\mathbf{x} \mapsto K(\mathbf{w},\mathbf{x})$$

*is real-analytic on $\mathbb{R}^d$, and in particular infinitely differentiable ($C^\infty$).*

*Proof.* Both $N(\mathbf{x}) = \langle\mathbf{w},\mathbf{x}\rangle^2$ and $D(\mathbf{x}) = \|\mathbf{w}-\mathbf{x}\|^2 + \epsilon$ are polynomials; since $D \geq \epsilon > 0$, $1/D$ is analytic and so is $K = N/D$. □

**Proposition C.11** (Lipschitz Continuity). *For any fixed $\mathbf{w} \in \mathbb{R}^d$ and $\epsilon > 0$, the kernel $K(\mathbf{w}, \mathbf{x})$ is globally Lipschitz continuous in $\mathbf{x}$.*

*Proof.* It suffices to show that $\sup_{\mathbf{x} \in \mathbb{R}^d} \|\nabla_{\mathbf{x}} K(\mathbf{w}, \mathbf{x})\| < \infty$.

Writing $K = N/D$ with $N(\mathbf{x}) = \langle \mathbf{w}, \mathbf{x} \rangle^2$ and $D(\mathbf{x}) = \|\mathbf{w} - \mathbf{x}\|^2 + \epsilon$, we compute

$$\nabla_{\mathbf{x}} K(\mathbf{w}, \mathbf{x}) = \frac{2\langle \mathbf{w}, \mathbf{x} \rangle \mathbf{w} \, D(\mathbf{x}) - 2\langle \mathbf{w}, \mathbf{x} \rangle^2 (\mathbf{x} - \mathbf{w})}{D(\mathbf{x})^2}.$$

Let $a = \|\mathbf{w}\|$, $r = \|\mathbf{x}\|$. By Cauchy–Schwarz,

$$|\langle \mathbf{w}, \mathbf{x} \rangle| \leq ar, \qquad \|\mathbf{x} - \mathbf{w}\| \leq r + a.$$

Moreover

$$D(\mathbf{x}) = \|\mathbf{w} - \mathbf{x}\|^2 + \epsilon \geq (r - a)^2 + \epsilon.$$

Substituting these bounds gives

$$\|\nabla_{\mathbf{x}} K(\mathbf{w}, \mathbf{x})\| \leq \frac{2a^2 r}{(r - a)^2 + \epsilon} + \frac{2a^2 r^2 (r + a)}{\left((r - a)^2 + \epsilon\right)^2} =: \Phi(r).$$

As $r \to \infty$, $\Phi(r) = O(1/r) \to 0$. Since $\Phi$ is continuous on $[0, \infty)$ and bounded at infinity, it attains a finite global maximum:

$$L(a, \epsilon) := \sup_{r \geq 0} \Phi(r) < \infty.$$

By the mean value theorem in $\mathbb{R}^d$,

$$|K(\mathbf{w}, \mathbf{x}) - K(\mathbf{w}, \mathbf{y})| \leq L(a, \epsilon) \|\mathbf{x} - \mathbf{y}\|,$$

so $K$ is globally Lipschitz in $\mathbf{x}$ with constant $L(a, \epsilon)$, depending on $\|\mathbf{w}\|$ and $\epsilon$. $\qquad\square$

### C.14 Universal Approximation Theorem for E-Product Networks

#### Setup and notation

Let $\mathcal{X} \subset \mathbb{R}^d$ be compact and fix $\varepsilon > 0$. Define the unit

$$g(x; w, b) := \frac{(x \cdot w + b)^2}{\|x - w\|^2 + \varepsilon}, \qquad w \in \mathbb{R}^d, \ b \in \mathbb{R},$$

and let

$$\mathcal{F} = \operatorname{span}\{ g(\cdot; w, b) \mid w \in \mathbb{R}^d, \ b \in \mathbb{R} \}.$$

Denote the inverse–multiquadric (IMQ) kernel centered at $w$ by

$$K(\cdot, w) := (\| \cdot - w\|^2 + \varepsilon)^{-1}.$$

**Lemma C.12** (Derivatives in the bias recover IMQ). *For every fixed $w \in \mathbb{R}^d$ and $b \in \mathbb{R}$ there holds, pointwise in $x$,*

$$\frac{\partial^2}{\partial b^2} g(x; w, b) = \frac{2}{\|x - w\|^2 + \varepsilon}.$$

*Moreover, for compact $\mathcal{X}$ the second derivative in $b$ is the uniform limit on $\mathcal{X}$ of finite difference quotients built from translates of $g$, and hence*

$$K(\cdot, w) = \tfrac{1}{2} \partial_b^2 g(\cdot; w, b) \in \overline{\mathcal{F}}$$

*(the uniform closure on $\mathcal{X}$).*

*Proof.* Fix $x, w$ and write

$$g(x; w, b) = \frac{(x \cdot w)^2 + 2b(x \cdot w) + b^2}{\|x - w\|^2 + \varepsilon}.$$

Differentiating in $b$ yields

$$\partial_b g(x; w, b) = \frac{2(x \cdot w) + 2b}{\|x - w\|^2 + \varepsilon}, \qquad \partial_b^2 g(x; w, b) = \frac{2}{\|x - w\|^2 + \varepsilon},$$

establishing the first identity.

Next, for each fixed $x \in \mathcal{X}$ the function $b \mapsto g(x; w, b)$ is a polynomial in $b$ whose denominator satisfies $\|x - w\|^2 + \varepsilon \geq \varepsilon > 0$. Hence $g$ is $C^\infty$ in $b$ and all derivatives $\partial_b^k g(\cdot; w, b)$ are continuous and uniformly bounded on $\mathcal{X}$. Thus finite difference quotients for $\partial_b$ and $\partial_b^2$ converge uniformly on $\mathcal{X}$ to the corresponding derivatives.

Each finite-difference quotient is a linear combination of finitely many translates $g(\cdot; w, b + h)$ and $g(\cdot; w, b)$, hence belongs to $\mathcal{F}$. Passing to the uniform limit shows that $\partial_b^2 g(\cdot; w, b) \in \overline{\mathcal{F}}$. Substituting the formula above yields

$$K(\cdot, w) = \tfrac{1}{2} \partial_b^2 g(\cdot; w, b) \in \overline{\mathcal{F}}.$$

$\square$

**Lemma C.13** (Fourier transform of IMQ and positivity). *Let $K(x) = (\|x\|^2 + \varepsilon)^{-1}$ on $\mathbb{R}^d$ with $\varepsilon > 0$. Its Fourier transform (distributional for $d \geq 4$; classical otherwise) is radial and equals*

$$\widehat{K}(\xi) = C_{d,\varepsilon} \|\xi\|^{1-d/2} K_{\frac{d}{2}-1}\big(\sqrt{\varepsilon}\, \|\xi\|\big), \tag{28}$$

*where $K_\nu$ is the modified Bessel function of the second kind and $C_{d,\varepsilon} > 0$. In particular, $\widehat{K}(\xi) > 0$ for all $\xi \neq 0$, with the removable value at $\xi = 0$ taken by continuity.*

*Proof.* Because $K$ is radial, its Fourier transform is radial and given by the Hankel transform formula

$$\widehat{K}(\xi) = (2\pi)^{d/2} \|\xi\|^{-(d/2-1)} \int_0^\infty r^{d/2} K(r) J_{d/2-1}(r\|\xi\|) \, dr.$$

Use the Laplace representation

$$K(r) = \frac{1}{r^2 + \varepsilon} = \int_0^\infty e^{-t(r^2 + \varepsilon)} \, dt,$$

valid by Tonelli since the integrand is positive. Substitute and switch integrals:

$$\widehat{K}(\xi) = (2\pi)^{d/2} \|\xi\|^{-(d/2-1)} \int_0^\infty e^{-t\varepsilon} \Big( \int_0^\infty r^{d/2} e^{-tr^2} J_{d/2-1}(r\|\xi\|) dr \Big) dt.$$

The inner integral is standard:

$$\int_0^\infty r^{\nu+1} e^{-tr^2} J_\nu(rs) \, dr = \frac{s^\nu}{(2t)^{\nu+1}} e^{-s^2/(4t)}, \qquad \nu > -1, \ t > 0.$$

Applying this with $\nu = d/2 - 1$ and simplifying gives an integral representation whose evaluation is the modified Bessel function $K_\nu$. The resulting constants combine to give (28) with $C_{d,\varepsilon} > 0$.

Since $K_\nu(z) > 0$ for all $z > 0$, we obtain $\widehat{K}(\xi) > 0$ for all $\xi \neq 0$. Continuity gives positivity at $\xi = 0$ as well. $\square$

**Theorem C.14** (Universal approximation for $\mathbb{E}$-product networks). *The class $\mathcal{F}$ is dense in $C(\mathcal{X})$ under the uniform norm.*

*Proof.* Lemma C.12 implies

$$K(\cdot, w) \in \overline{\mathcal{F}} \qquad \text{for every } w \in \mathbb{R}^d.$$

Hence

$$\text{span}\{K(\cdot, w) : w \in \mathbb{R}^d\} \subseteq \overline{\mathcal{F}}.$$

Let $\mu$ be a finite signed Borel measure supported on $\mathcal{X}$ such that

$$\int_{\mathcal{X}} K(x, w)\, d\mu(x) = 0 \qquad \forall w \in \mathbb{R}^d.$$

Extend $\mu$ by 0 outside $\mathcal{X}$ to obtain a compactly supported measure on $\mathbb{R}^d$. The integral condition is equivalent to

$$(K * \mu)(w) = 0, \qquad \forall w \in \mathbb{R}^d.$$

Taking Fourier transforms gives

$$\widehat{K}(\xi)\, \widehat{\mu}(\xi) = 0.$$

By Lemma C.13, $\widehat{K}(\xi) > 0$ for all $\xi$, hence $\widehat{\mu}(\xi) \equiv 0$. Uniqueness of Fourier transform for compactly supported finite measures implies $\mu \equiv 0$.

By the Hahn–Banach / Riesz duality criterion, the span of $\{K(\cdot, w) : w \in \mathbb{R}^d\}$ is dense in $C(\mathcal{X})$. Since this span is contained in $\overline{\mathcal{F}}$, we conclude $\overline{\mathcal{F}} = C(\mathcal{X})$. $\qquad\square$

**Corollary C.15** (Practical corollary: single-hidden-layer networks are universal). *Let networks of the form*

$$x \mapsto \sum_{i=1}^{n} \alpha_i\, g(x; w_i, b_i) + c$$

*be considered. Then for every continuous $f \in C(\mathcal{X})$ and every $\delta > 0$ there exist $n$ and parameters $\{\alpha_i, w_i, b_i, c\}$ such that*

$$\sup_{x \in \mathcal{X}} \left| f(x) - \sum_{i=1}^{n} \alpha_i g(x; w_i, b_i) - c \right| < \delta.$$

*Proof.* Immediate from Theorem C.14. $\qquad\square$

### C.15 Information-Geometric Foundations of the $\mathbb{E}$-Product

#### C.15.1 Definition and Geometric Interpretation

We consider probability distributions in the simplex $\Delta^{n-1} = \{\mathbf{p} \in \mathbb{R}^n_{\geq 0} : \sum_{i=1}^{n} p_i = 1\}$. While information geometry traditionally employs the Fisher metric, we establish a novel connection to Euclidean geometry through the $\mathbb{E}$-product.

**Definition C.1** ($\mathbb{E}$-Product: Geometric Similarity Measure). *For distinct distributions $\mathbf{p}, \mathbf{q} \in \Delta^{n-1}$, the $\mathbb{E}$-product is defined as:*

$$\mathbb{E}(\mathbf{p}, \mathbf{q}) := \frac{(\mathbf{p} \cdot \mathbf{q})^2}{\|\mathbf{p} - \mathbf{q}\|_2^2}$$

*where:*

- $\mathbf{p} \cdot \mathbf{q} = \sum_{i=1}^{n} p_i q_i$ *measures distributional alignment*

- $\|\mathbf{p} - \mathbf{q}\|_2^2 = \sum_{i=1}^{n} (p_i - q_i)^2$ *quantifies Euclidean dissimilarity*

*This ratio captures the tension between distributional agreement and geometric separation.*

**Remark C.16** (Singularity and Invariance Properties)**.** *When* $\mathbf{p} = \mathbf{q}$, *we define* $\mathbf{E}(\mathbf{p}, \mathbf{q}) := \infty$ *via the limit:*

$$\lim_{\mathbf{q} \to \mathbf{p}} \mathbf{E}(\mathbf{p}, \mathbf{q}) = \infty$$

*reflecting maximal self-similarity. The* $\mathbf{E}$*-product exhibits two key properties:*

1. ***Symmetry:*** $\mathbf{E}(\mathbf{p}, \mathbf{q}) = \mathbf{E}(\mathbf{q}, \mathbf{p})$

2. ***Scale Invariance:*** *Invariant under index permutation*

### C.15.2 EXTREMAL SIMILARITY THEOREMS

We collect here the detailed proofs of the extremal similarity results stated in the main text.

**Proof of Minimal Similarity and Statistical Orthogonality.**

*Proof.* ($\Rightarrow$) Assume $\mathbf{E}(\mathbf{p}, \mathbf{q}) = 0$. Since $\mathbf{p} \neq \mathbf{q}$, $\|\mathbf{p} - \mathbf{q}\|_2^2 > 0$. Thus $(\mathbf{p} \cdot \mathbf{q})^2 = 0 \Rightarrow \sum p_i q_i = 0$. By non-negativity of probabilities, $p_i q_i = 0 \; \forall i$, hence $\mathrm{supp}(\mathbf{p}) \cap \mathrm{supp}(\mathbf{q}) = \emptyset$.

($\Leftarrow$) Disjoint supports imply $\forall i : (p_i > 0 \Rightarrow q_i = 0)$ and vice versa. Thus $\mathbf{p} \cdot \mathbf{q} = 0$, so $\mathbf{E}(\mathbf{p}, \mathbf{q}) = 0$.

The KL divergence $\mathrm{KL}(\mathbf{p} \| \mathbf{q})$ contains terms $\log(p_i/q_i)$ where $p_i > 0$ and $q_i = 0$, causing divergence. Similar reasoning applies to $\mathrm{KL}(\mathbf{q} \| \mathbf{p})$ and cross-entropy $H(\mathbf{p}, \mathbf{q})$(Cover & Thomas, 2006). $\qquad\square$

**Proof of Maximal Similarity and Distributional Identity.**

*Proof.* ($\Rightarrow$) Suppose $\mathbf{E}(\mathbf{p}, \mathbf{q}) \to \infty$. By Cauchy-Schwarz(Horn & Johnson, 2012), $\mathbf{p} \cdot \mathbf{q} \leq \|\mathbf{p}\|_2 \|\mathbf{q}\|_2 \leq 1$. Since the numerator is bounded, $\|\mathbf{p} - \mathbf{q}\|_2^2 \to 0$, implying $\mathbf{p} = \mathbf{q}$.

($\Leftarrow$) For $\mathbf{p} = \mathbf{q}$, consider $\mathbf{q}^{(k)} \to \mathbf{p}$. Then:

$$\mathbf{p} \cdot \mathbf{q}^{(k)} \to \|\mathbf{p}\|_2^2 \geq \tfrac{1}{n} > 0 \quad (\text{since } \|\mathbf{p}\|_2^2 \geq \tfrac{1}{n} \text{ by Cauchy-Schwarz})$$

while $\|\mathbf{p} - \mathbf{q}^{(k)}\|_2^2 \to 0$, so $\mathbf{E}(\mathbf{p}, \mathbf{q}^{(k)}) \to \infty$.

When $\mathbf{p} = \mathbf{q}$, $\log(p_i/q_i) = 0$ for all $i$, so $\mathrm{KL}(\mathbf{p} \| \mathbf{q}) = 0$. Cross-entropy reduces to entropy when distributions are identical. $\qquad\square$

**Remark C.17** (Duality of Orthogonality Concepts)**.** *The* $\mathbf{E}$*-product unifies three distinct notions of orthogonality:*

$$\textit{Euclidean: } \mathbf{p} \perp \mathbf{q} \iff \mathbf{p} \cdot \mathbf{q} = 0$$
$$\textit{Combinatorial: } \mathrm{supp}(\mathbf{p}) \cap \mathrm{supp}(\mathbf{q}) = \emptyset$$
$$\textit{Information-Theoretic: } \mathrm{KL}(\mathbf{p} \| \mathbf{q}) = \infty$$

*Theorem 2.2 establishes their equivalence through* $\mathbf{E}(\mathbf{p}, \mathbf{q}) = 0$*. This contrasts with Fisher-based orthogonality, which depends on manifold curvature.*

**Remark C.18** (Geometric-Information Duality)**.** *The* $\mathbf{E}$*-product creates a bridge between geometric and probabilistic perspectives:*

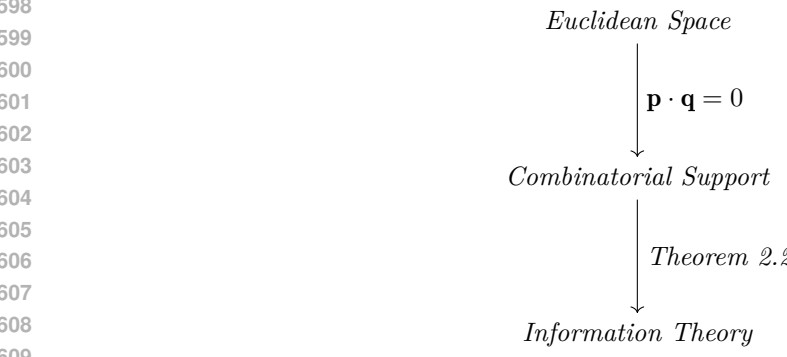

# D    DETAILED XOR ANALYSIS AND GRADIENT PROPERTIES

This section provides a comprehensive analysis of the Ɛ-product's ability to solve the XOR problem and its gradient properties that enable stable learning.

## D.1    MATHEMATICAL FOUNDATION FOR XOR SOLUTION

The Ɛ-product's non-linearity is not merely a mathematical curiosity; it has practical implications for neural computation. By integrating alignment and proximity into a single operator, the Ɛ-product allows for more nuanced feature learning. It can adaptively respond to inputs based on their geometric relationships with learned weight vectors, enabling the network to capture complex patterns without the need for separate activation functions.

The XOR problem is not linearly separable and thus cannot be solved by a single traditional neuron (linear perceptron). However, a single Ɛ-product unit can solve this problem due to its inherent non-linearity. We have formally proven that the Ɛ-product is a valid Mercer kernel (Theorem 2.1; see Appendix C.9)(Mercer, 1909; Hofmann et al., 2008; Micchelli et al., 2006; Schölkopf et al., 1997; Cortes, 1995; Jacot et al., 2018).

## D.2    GRADIENT ANALYSIS AND LEARNING STABILITY

To understand the Ɛ-product's behavior during learning, we analyze its gradient properties. A key property for stable training is that the gradient with respect to the input, $\nabla_{\mathbf{x}}\mathcal{K}_{\text{Ɛ}}$, diminishes as the input $\mathbf{x}$ moves far from the weight vector $\mathbf{w}$. This ensures that distant outliers do not cause large, destabilizing updates. We have formally proven this as a stability Proposition (Proposition C.9; Appendix C.12), demonstrating that $\lim_{\|\mathbf{x}\|\to\infty}\|\nabla_{\mathbf{x}}\mathcal{K}_{\text{Ɛ}}(\mathbf{w},\mathbf{x})\| = 0$.

The presence of $\epsilon$ in the denominator ensures that the derivative remains well-defined, avoiding division by zero and contributing to numerical stability. This contrasts with activation functions like ReLU, which have a derivative of zero for negative inputs, potentially leading to "dead neurons." The smooth and generally non-zero gradient of the Ɛ-product is hypothesized to contribute to more stable and efficient learning dynamics, reducing the reliance on auxiliary mechanisms like complex normalization schemes.

The non-linearity is thus not an add-on but an intrinsic property derived from the direct mathematical interaction of vector projection (alignment, via the $(\mathbf{w}^{\top}\mathbf{x})^2$ term) and vector distance (proximity, via the $\|\mathbf{w} - \mathbf{x}\|^2 + \epsilon$ term). This provides a mathematically grounded basis for feature learning, as the unit becomes selectively responsive to inputs that exhibit specific geometric relationships, both

in terms of angular alignment and spatial proximity, to its learned weight vector $\mathbf{w}$. Consequently, $\mathbf{w}$ can be interpreted as a learned feature template or prototype that the unit is tuned to detect, with the ⊵-product quantifying the degree of match in a nuanced, non-linear fashion.

### D.3 OPTIMIZATION LANDSCAPE ANALYSIS

The gradient of the ⊵-product, being responsive across the input space, actively pushes the neuron's weights away from configurations that would lead to a zero output (neuron death, e.g., at an input of $[0, 0]$ for this problem if weights were also near zero). This contrasts with a simple dot product neuron where the gradient might vanish or lead to a global minimum at zero output for certain problems.

For instance, when considering gradient-based optimization, the loss landscape "seen" by the ⊵-product neuron in the XOR context would exhibit a peak or high loss at $[0, 0]$ (if that were the target for non-zero outputs), encouraging weights to move towards a state that correctly classifies. Conversely, a simple dot product neuron might present a loss landscape where a gradient-based optimizer could find a stable (but incorrect) minimum at zero output. This ability to avoid such dead zones and actively shape the decision boundary makes it helpful to solve problems like XOR with a single unit, leveraging its inherent non-linearity as a mathematical kernel.

### D.4 GEOMETRIC INTERPRETATION OF DECISION BOUNDARIES

Conceptually, the decision boundary or vector field generated by a simple dot product neuron is linear, forming a hyperplane that attempts to separate data points. In contrast, the ⊵-product generates a more complex, non-linear vector field. This field can be visualized as creating a series of potential wells or peaks centered around the weight vector $\mathbf{w}$, with the strength of influence decaying with distance. This structure allows for more nuanced and localized responses. The ⊵-product solves XOR through its Localized Quadratic Expressivity. The quadratic numerator $(\mathbf{w}^\top \mathbf{x})^2$ provides the necessary non-linearity, exhibiting a Superposition property that manifests as Sign Invariance: the response is identical for $\mathbf{x}$ and $-\mathbf{x}$. This allows the network to classify antipodal points similarly. Geometrically, the solution corresponds to an orthogonal valley—the region where $\mathbf{w}^\top \mathbf{x} \approx 0$—where the loss is minimal, effectively separating the XOR classes.

Figure 5: Comparison of loss landscapes: dot-product neuron (left) exhibits spurious minima leading to vanishing gradients, while ⊵-product neuron (right) yields non-degenerate gradients enabling optimization to reach correct separators.

## E DETAILED VORTEX DYNAMICS ANALYSIS

This section provides a comprehensive mathematical analysis of the vortex-like learning dynamics exhibited by ⊵-product neurons and their space-partitioning behavior.

### E.1 FUNDAMENTAL LEARNING DYNAMICS

We begin by analyzing the fundamental learning dynamics that emerge in both conventional and our proposed architectures. In artificial intelligence, competitive learning manifests in various forms, whether through linear classification using dot products or clustering using Euclidean distances. Both approaches involve partitioning the feature space between neurons, which can be conceptu-

alized as prototype learning where each neuron claims a territorial "field" in the representation space.

### E.2 Conventional vs. $\mathbb{E}$-Product Decision Boundaries

In a conventional linear model, the logit for each class $i$ is computed as:

$$z_i = \mathbf{w}_i^T \mathbf{x} \tag{29}$$

where $\mathbf{w}_i$ is the weight vector (prototype) for class $i$, and $\mathbf{x}$ is the input vector. The softmax function then normalizes these logits into probabilities:

$$p_i = \frac{\exp(z_i)}{\sum_{j=1}^{C} \exp(z_j)} \tag{30}$$

The decision boundary between any two classes $i$ and $j$ forms a linear hyperplane defined by:

$$(\mathbf{w}_i - \mathbf{w}_j)^T \mathbf{x} = 0 \tag{31}$$

During training via gradient descent, each prototype $\mathbf{w}_i$ is updated to maximize its alignment with the data distributions of its assigned class. This optimization process often leads to an unbounded increase in prototype magnitudes, as $\|\mathbf{w}_i\| \to \infty$ directly amplifies the logit $z_i$, thereby increasing the model's confidence. However, the decision boundaries themselves remain linear hyperplanes, creating rigid geometric separations in the feature space.

In contrast, the non-linear $\mathbb{E}$-product allows neurons to learn more representative prototypes for each class, leading to the formation of more nuanced decision boundaries. For the $\mathbb{E}$-product, the response of neuron $i$ to input $\mathbf{x}$ is given by:

$$z_i = \mathbb{E}(\mathbf{w}_i, \mathbf{x}) = \frac{\langle \mathbf{w}_i, \mathbf{x} \rangle^2}{\|\mathbf{w}_i - \mathbf{x}\|^2 + \epsilon} \tag{32}$$

This formulation embodies the signal-to-noise ratio interpretation established in our theoretical framework (Appendix C.15), where the squared dot product $\langle \mathbf{w}_i, \mathbf{x} \rangle^2$ represents the "signal" of distributional alignment, and $\|\mathbf{w}_i - \mathbf{x}\|^2$ quantifies the "noise" of dissimilarity.

### E.3 Softmax Dynamics and Competitive Learning

Similarly to conventional neurons, the $\mathbb{E}$-product outputs are normalized using the softmax function:

$$p_i = \frac{\exp(z_i)}{\sum_{j=1}^{C} \exp(z_j)} = \frac{\exp\left(\frac{\langle \mathbf{w}_i, \mathbf{x} \rangle^2}{\|\mathbf{w}_i - \mathbf{x}\|^2 + \epsilon}\right)}{\sum_{j=1}^{C} \exp\left(\frac{\langle \mathbf{w}_j, \mathbf{x} \rangle^2}{\|\mathbf{w}_j - \mathbf{x}\|^2 + \epsilon}\right)} \tag{33}$$

This softmax normalization serves a crucial role in the competitive dynamics of $\mathbb{E}$-product neurons. The softmax function acts as a transformation that maps from the real-valued $\mathbb{E}$-product responses $z_i \in \mathbb{R}$ to a delta distribution $\delta_i$ in probability space. This softmax distribution over $\mathbb{E}$-product scores can be interpreted as the *posterior responsibility* of each prototype (neuron) for the input, drawing a direct connection to Gaussian Mixture Models (GMMs) and expectation-maximization frameworks.

The softmax can also be viewed as computing a categorical distribution proportional to exponentiated log-likelihoods, which in this case derive from a geometric $\mathbb{E}$-product similarity rather than

traditional probabilistic assumptions. This bridges the gap between probabilistic views (such as EM algorithms and classification) and our geometric formulation, providing a principled foundation for the competitive dynamics.

As training progresses and the differences between $\mathsf{E}$-product responses become more pronounced, the softmax transformation approaches a delta distribution, where the winning neuron (with the highest $\mathsf{E}$-product response) approaches probability 1 while all others approach 0. This yields competitive learning dynamics in which neurons specialize on distinct regions of the input space.

### E.4 MATHEMATICAL CHARACTERIZATION OF DECISION BOUNDARIES

The decision boundary between two neurons with prototypes $\mathbf{w}_i$ and $\mathbf{w}_j$ is defined by the condition where their responses are equal:

$$\mathsf{E}(\mathbf{w}_i, \mathbf{x}) = \mathsf{E}(\mathbf{w}_j, \mathbf{x}) \tag{34}$$

Expanding this condition:

$$\frac{\langle \mathbf{w}_i, \mathbf{x} \rangle^2}{\|\mathbf{w}_i - \mathbf{x}\|^2 + \epsilon} = \frac{\langle \mathbf{w}_j, \mathbf{x} \rangle^2}{\|\mathbf{w}_j - \mathbf{x}\|^2 + \epsilon} \tag{35}$$

Cross-multiplying and rearranging:

$$\langle \mathbf{w}_i, \mathbf{x} \rangle^2 (\|\mathbf{w}_j - \mathbf{x}\|^2 + \epsilon) = \langle \mathbf{w}_j, \mathbf{x} \rangle^2 (\|\mathbf{w}_i - \mathbf{x}\|^2 + \epsilon) \tag{36}$$

This equation defines a non-linear decision boundary that depends on both alignment (via squared dot products) and proximity (via squared distances) between the input and each prototype. Unlike the linear hyperplane formed by conventional dot-product neurons, the $\mathsf{E}$-product induces curved algebraic decision surfaces (analogous to level sets of a potential).

### E.5 SPACE-PARTITIONING PROPERTIES

The space-partitioning behavior of the $\mathsf{E}$-product exhibits several key properties:

- **Distance Penalization and Locality**: The factor $(\|\mathbf{w} - \mathbf{x}\|^2 + \epsilon)^{-1}$ enforces spatial selectivity. Along rays $\mathbf{x} = k\mathbf{u}$ with $\|\mathbf{u}\| = 1$, one has $\lim_{k \to \infty} \mathsf{E}(\mathbf{w}, k\mathbf{u}) = (\mathbf{w} \cdot \mathbf{u})^2 = \|\mathbf{w}\|^2 \cos^2 \theta$, so responses are largest near $\mathbf{x} \approx \mathbf{w}$ and remain bounded far away (cf. Proposition C.9 for the gradient decay). In physical terms, this is analogous to an inverse-square distance penalty.

- **Alignment Weighting**: The numerator is $(\mathbf{w}^\top \mathbf{x})^2$, so $\mathsf{E}(\mathbf{w}, \mathbf{x}) = 0$ if and only if $\mathbf{w} \perp \mathbf{x}$, and high values require strong alignment (large $|\cos \theta|$) in addition to proximity.

- **Global Boundedness for Fixed $\mathbf{w}$**: For $\epsilon > 0$ and fixed $\mathbf{w}$, $0 \le \mathsf{E}(\mathbf{w}, \mathbf{x}) \le \|\mathbf{w}\|^4/\epsilon$ for all $\mathbf{x}$, with $\mathsf{E}(\mathbf{w}, \mathbf{w}) = \|\mathbf{w}\|^4/\epsilon$ and $\limsup_{\|\mathbf{x}\| \to \infty} \mathsf{E}(\mathbf{w}, \mathbf{x}) \le \|\mathbf{w}\|^2$. Thus each unit defines a bounded activation landscape (cf. Proposition C.6).

- **Nonlinear Decision Regions**: The pairwise decision boundary between units $i$ and $j$ is the algebraic surface

$$\langle \mathbf{w}_i, \mathbf{x} \rangle^2 (\|\mathbf{w}_j - \mathbf{x}\|^2 + \epsilon) - \langle \mathbf{w}_j, \mathbf{x} \rangle^2 (\|\mathbf{w}_i - \mathbf{x}\|^2 + \epsilon) = 0,$$

which is generically non-linear and induces curved class regions. (Geometrically, one may view these as level sets akin to equipotential surfaces.)

### E.6 Vortex Field Dynamics

This vortex phenomenon allows each $\boxminus$-product neuron to create a territorial "field" in the representation space, where data points are pulled toward the dominant prototype based on both similarity and proximity metrics. The field each neuron occupies can indeed be considered a vortex, where the strength of attraction follows an inverse-square law, creating more natural and geometrically faithful decision boundaries.

The combination of the $\boxminus$-product's vortex-like attraction and the softmax's competitive normalization creates a powerful space partitioning mechanism. Each neuron's vortex field competes with others through the softmax transformation, and the neuron with the strongest local attraction (highest $\boxminus$-product response) wins that region. Over time, this leads to a natural tessellation of the input space, where each neuron's territory is defined by the regions where its vortex field dominates. The softmax transformation $\mathbb{R}^C \to \Delta^{C-1}$ (where $\Delta^{C-1}$ is the $(C-1)$-dimensional probability simplex) ensures that these territorial boundaries are sharp and well-defined, transforming the continuous real-valued responses into discrete delta distributions that clearly assign each input to its dominant neuron.

### E.7 Orthogonality and Competitive Dynamics

The competitive learning behavior observed in practice is theoretically grounded in our Orthogonality-Entropy Connection. When two prototypes $\mathbf{w}_i$ and $\mathbf{w}_j$ develop disjoint support regions, they become Euclidean orthogonal ($\mathbf{w}_i \perp \mathbf{w}_j$), which corresponds to:

$$\boxminus(\mathbf{w}_i, \mathbf{w}_j) = 0 \quad \text{and} \quad H(\mathbf{w}_i, \mathbf{w}_j) = \infty \tag{37}$$

This geometric-probabilistic duality explains why neurons naturally develop specialized, non-overlapping representations during competitive learning. The infinite cross-entropy between orthogonal prototypes creates strong pressure for territorial separation, preventing the collapse to identical representations that can plague conventional competitive learning systems.

These prototypes are optimized to maximize parallelism and minimize distance to all points within their class distribution. When minimizing distance becomes challenging, the properties of the $\boxminus$-product enable the prototype to exist in a superposition state, prioritizing the maximization of parallelism over strict distance minimization.

### E.8 MNIST Prototype Analysis: Detailed Mathematical Framework

#### E.8.1 Network Architecture and Dimensional Structure

In our MNIST experiments, the network consists of $C = 10$ neurons, each corresponding to one of the digit classes (0–9). Each neuron's prototype is represented as a vector $\mathbf{w}_i \in \mathbb{R}^{784}$, where $i = 1, \ldots, 10$. The input images $\mathbf{x} \in \mathbb{R}^{784}$ are obtained by flattening the original $28 \times 28$ pixel images, so each neuron's prototype $\mathbf{w}_i$ has the same dimensionality as the input, i.e., $\mathbf{w}_i = (w_{i,1}, w_{i,2}, \ldots, w_{i,784})$. This structure allows each neuron to learn class-specific features in the full image space.

#### E.8.2 Prototype Evolution Dynamics

The prototype evolution during training reveals fundamental differences between conventional unbounded growth and bounded $\boxminus$-responses. Our experiments on the full MNIST dataset (60,000 training samples, 10,000 test samples) demonstrate that both models achieve competitive perfor-

mance: the linear classifier reaches 92.08% test accuracy while the E-product classifier achieves 92.38% test accuracy after 5 epochs of training with the Adam optimizer (learning rate 0.001).

Quantitatively, the linear prototypes exhibit unbounded growth, with the mean magnitude increasing by 13.8% (from 1.58 to 1.80) during training. In contrast, the E-product prototypes show a slight contraction of 4.5% (from 5.02 to 4.79), confirming the bounded nature of the learned representations.

Figure 6 visualizes the learned prototypes for both models. The conventional linear model produces prototypes that exhibit unbounded growth—these prototypes become increasingly diffuse and less interpretable as they grow to maximize margin separation. The resulting digit representations are blurry and lack the fine-grained features necessary for robust classification.

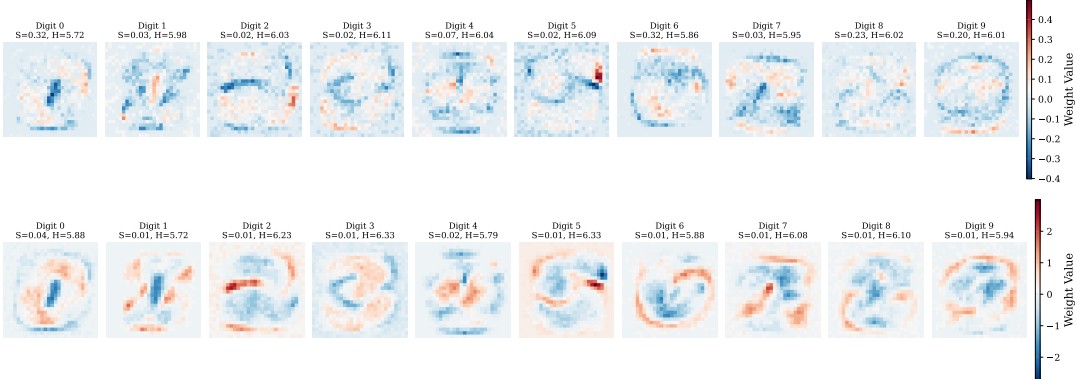

Figure 6: Learned prototypes for Linear (top) and E-product (bottom) classifiers. The linear prototypes show unbounded growth and noise, while E-product prototypes exhibit localized, sharp features with clear digit structure, consistent with bounded response fields.

In contrast, the E-product method produces prototypes that exemplify bounded response fields as predicted by our theoretical framework. Each digit prototype exhibits three key characteristics:

**Localized Concentration**   Sharp, well-defined features that correspond to the bounded potential wells predicted by our Minimal and Maximal Similarity Characterizations (Theorems 2.2 and 2.3). This localization emerges from the E-product's inherent bounded response field, which prevents the unbounded growth characteristic of linear neurons.

**Class-Specific Territorial Structure**   Each prototype captures unique digit characteristics, reflecting competitive territorial dynamics where each neuron dominates specific regions of the input space. This territorial behavior arises from the vortex-like dynamics of the E-product, creating natural tessellation of the feature space.

**Geometric Fidelity**   The prototypes maintain geometric coherence with actual digit structure, confirming that the signal-to-noise ratio optimization preserves meaningful visual patterns. This preservation is a direct consequence of the E-product's ability to balance alignment and proximity measures.

### E.8.3 Learnable Scaling Factor Dynamics

A critical component of the Æ-product classifier is the learnable scaling factor $\alpha$, which modulates the magnitude of the Æ-product scores. In our implementation, this factor is initialized to $\alpha_0 = 1.0$ and learned jointly with the prototypes during training.

Figure 7 shows the training dynamics, including the evolution of the scaling factor. Over 5 epochs, the scaling factor increases steadily from 1.0 to approximately 2.68, representing a 168% increase. This growth pattern reveals that the model learns to amplify the Æ-product scores to achieve better separation in the logit space, compensating for the inherently bounded nature of the Æ-product response.

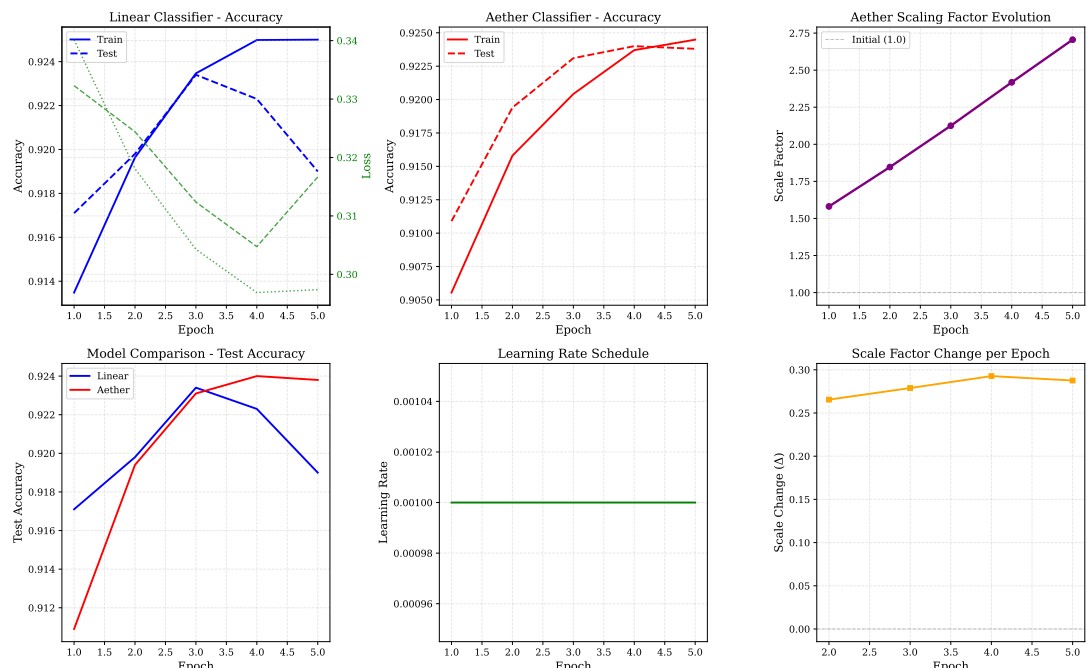

Figure 7: Training dynamics for MNIST classification. **Top row:** Accuracy curves for linear (left) and Æ-product (center) classifiers, with Æ-product scaling factor evolution (right). **Bottom row:** Model comparison (left), learning rate schedule (center), and per-epoch scale change (right). The scaling factor increases from 1.0 to 2.68 over training, indicating the model learns to amplify Æ-product scores for better classification.

The scaling factor evolution can be understood through the lens of optimization dynamics. The Æ-product naturally produces bounded responses due to its ratio structure, with values typically in the range [0, 10] depending on the alignment and distance between inputs and prototypes. By learning to scale these responses up, the model increases the dynamic range of the logits, which improves the discriminative power of the softmax layer. This adaptive scaling mechanism is analogous to a temperature parameter in softmax, but learned end-to-end rather than hand-tuned.

### E.8.4 Superposition and Prototype Inversion: Mathematical Analysis

A unique property of the E̅-product neuron is its ability to exist in a superposition state, which can be empirically demonstrated by inverting the learned prototype. This phenomenon provides insight into the fundamental differences between conventional and E̅-product neurons.

**Conventional Dot Product Behavior**  For a conventional dot product neuron, if $\mathbf{w}$ is a learned prototype, replacing $\mathbf{w}$ with $-\mathbf{w}$ (i.e., multiplying by $-1$) at test time flips the sign of the logit:

$$z = \mathbf{w}^T\mathbf{x} \quad \longrightarrow \quad z' = (-\mathbf{w})^T\mathbf{x} = -z \tag{38}$$

This sign flip causes the softmax output to assign high probability to the incorrect class, resulting in a catastrophic drop in accuracy. In our experiments, the linear classifier's accuracy drops from 92.04% to 0.01% (near complete failure) when all weight vectors are inverted.

**E̅-Product Inversion Dynamics**  For the E̅-product neuron, the response is:

$$z = \mathbb{E}(\mathbf{w}, \mathbf{x}) = \frac{(\mathbf{w}^T\mathbf{x})^2}{\|\mathbf{w} - \mathbf{x}\|^2 + \epsilon} \tag{39}$$

Multiplying $\mathbf{w}$ by $-1$ leaves the numerator unchanged, since $(-\mathbf{w})^T\mathbf{x} = -\mathbf{w}^T\mathbf{x}$ and $(-\mathbf{w}^T\mathbf{x})^2 = (\mathbf{w}^T\mathbf{x})^2$. The denominator, however, changes from $\|\mathbf{w} - \mathbf{x}\|^2$ to $\|\mathbf{w} + \mathbf{x}\|^2$:

$$\|\mathbf{w} + \mathbf{x}\|^2 = (\mathbf{w} + \mathbf{x})^T(\mathbf{w} + \mathbf{x}) \tag{40}$$

$$= \|\mathbf{w}\|^2 + 2\mathbf{w}^T\mathbf{x} + \|\mathbf{x}\|^2 \tag{41}$$

While exact invariance is not guaranteed due to this denominator change, our experiments confirm substantial robustness to prototype inversion. Figure 8 shows the comparison: when all prototypes are inverted (multiplied by $-1$), the linear classifier's accuracy drops from 92.04% to 0.01% (complete catastrophic failure), while the E̅-product classifier maintains 84.67% accuracy, experiencing only a 7.82% degradation. This remarkable robustness stems from the squared numerator term, which makes the E̅-product inherently invariant to the sign of the dot product and thus less sensitive to prototype orientation than linear neurons.

**Superposition Interpretation**  This property allows the E̅-product neuron to yield two valid solutions to the same input without retraining, a phenomenon not observed in conventional dot product neurons. The mathematical basis for this superposition lies in the E̅-product's non-linear structure, which creates multiple local optima that can represent the same classification boundary through different prototype orientations.

### E.8.5 Prototype Orthogonality Analysis

To further understand the learned representations, we analyze the pairwise orthogonality between prototypes. Figure 9 shows the orthogonality matrices for both models. For the linear classifier, we compute the orthogonality from standard dot products between weight vectors. For the E̅-product classifier, we compute orthogonality derived from the log-E̅-product between prototypes.

The E̅similarity matrices reveal fundamentally different prototype organization strategies. The linear classifier exhibits highly orthogonal prototypes with a mean log-E̅ similarity of only 0.0325 and a narrow range (max 0.23), indicating that the linear model learns nearly independent weight vectors that span complementary subspaces. This strong orthogonalization is a consequence of the

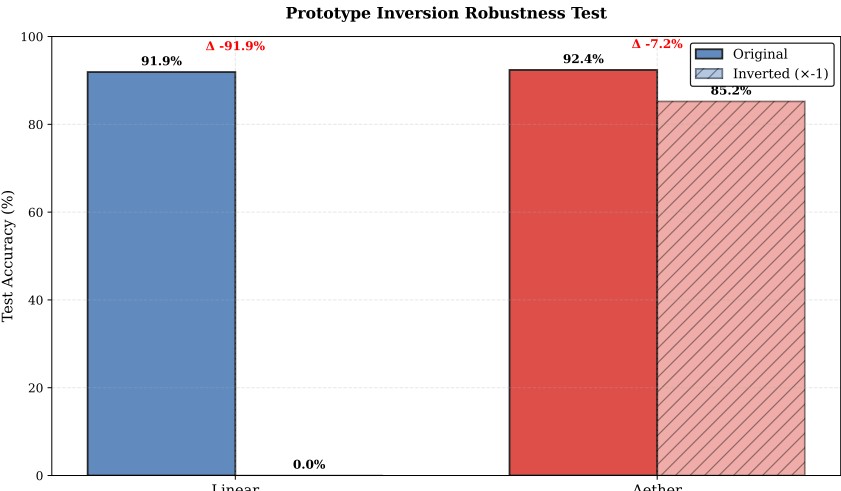

Figure 8: Prototype inversion experiment comparing linear and $\mathbf{E}$-product classifiers. Both models are trained normally, then all prototypes/weights are multiplied by $-1$ at test time. The linear classifier suffers catastrophic failure ($92.29\% \rightarrow \sim 0\%$), while the $\mathbf{E}$-product classifier shows robustness due to its squared numerator term.

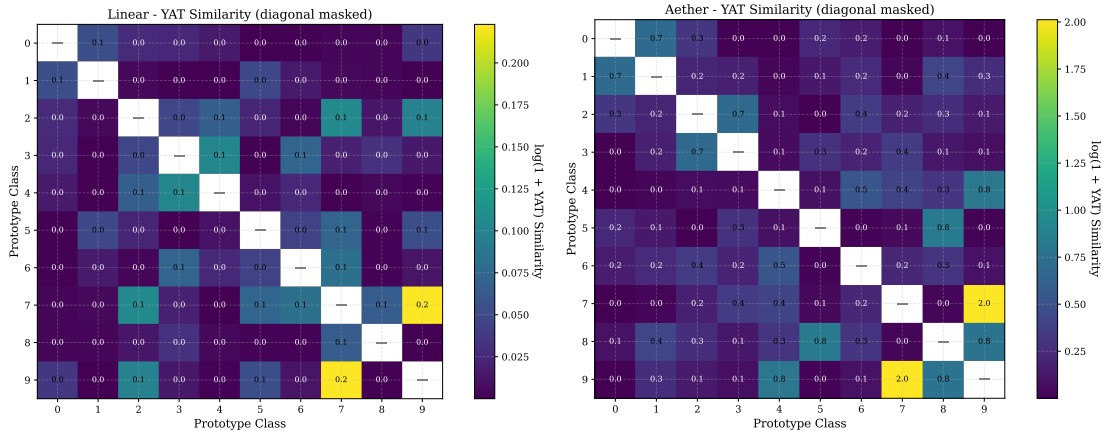

Figure 9: Pairwise orthogonality matrices between learned prototypes. **Left:** Orthogonality for linear classifier weights. **Right:** Orthogonality for $\mathbf{E}$-product prototypes. The $\mathbf{E}$-product prototypes exhibit stronger orthogonality (higher dynamic range), suggesting better class separation in the prototype space.

unbounded dot product's tendency to maximize margin separation through perpendicular decision boundaries.

In stark contrast, the Ɇ-product prototypes show dramatically higher mean similarity (0.274, over 8× larger) with an exceptionally wide dynamic range spanning from near-zero (0.0005) to 2.01—nearly 9× the linear maximum. This reveals a qualitatively different organizational principle: rather than enforcing global orthogonality, the Ɇ-product creates a *heterogeneous territorial structure* where some prototype pairs exhibit strong local correlations (e.g., classes 7–9 with similarity 2.01, and classes 4–9 with 0.79) while others remain effectively decoupled (e.g., classes 0–3 with 0.0005). This selective coupling pattern is consistent with the vortex-like dynamics of our theoretical framework, where prototypes carve out overlapping response regions in high-correlation areas yet maintain sharp boundaries where discrimination is critical, allowing the model to capture subtle intra-class variations while preserving inter-class separation.

### E.8.6 CONFUSION MATRIX ANALYSIS

We evaluate classification robustness through the confusion matrices presented in Figure 10. The quantitative difference between the architectures is evident in the structure of error distribution:

**Diagonal Dominance and Error Suppression** The Ɇ-product classifier exhibits sharper diagonalization compared to the linear baseline, achieving a higher overall accuracy of 92.38% versus 91.90% for the linear model. While the linear model shows diffuse off-diagonal elements indicating "leakage" between morphologically similar digits, the Ɇ-product minimizes these inter-class ambiguities. For instance, the linear model misclassifies 57 instances of digit '2' as '8', whereas the Ɇ-product reduces this to 38, a 33% reduction in error. Similarly, for the confusion pair $7 \rightarrow 9$, the linear model makes 42 errors compared to 30 for the Ɇ-product. This suppression of off-diagonal noise is a direct consequence of the *competitive territorial dynamics*: the inverse-square decay in the Ɇ-product creates steeper decision gradients at the boundaries between classes.

**Robustness to Ambiguity** The linear classifier's confusion patterns reflect its reliance on global hyperplane separation, which forces a trade-off between maximizing margins for easy examples and correctly classifying boundary cases. This is evident in the misclassification of digit '4' as '9', where the linear model fails on 40 samples, while the Ɇ-product improves this to 29 errors. The Ɇ-product's confusion matrix demonstrates that the model maintains high precision even for boundary cases, particularly on harder classes like 2, 3, and 7, confirming that the learned prototypes act as local attractors that naturally reject inconsistent inputs.

### E.8.7 WEIGHT DISTRIBUTION AND SPARSITY ANALYSIS

The statistical properties of the learned parameters, summarized in Figure 11, reveal a fundamental divergence in how the two models encode information.

**Magnitude Stability vs. Unbounded Growth** A critical distinction lies in the stability of the weight magnitudes. The Ɇ-product prototypes settle into a high-magnitude stable state ($\mu = 4.79, \sigma = 0.38$), exhibiting a slight contraction of 4.5% during the final training phase. Conversely, the linear weights ($\mu = 1.80, \sigma = 0.22$) exhibit characteristic unbounded growth (increasing by 13.8%) as they attempt to maximize the dot-product margin. This confirms that the Ɇ-product optimization landscape contains stable minima (bounded potential wells), whereas the linear landscape promotes indefinite norm growth.

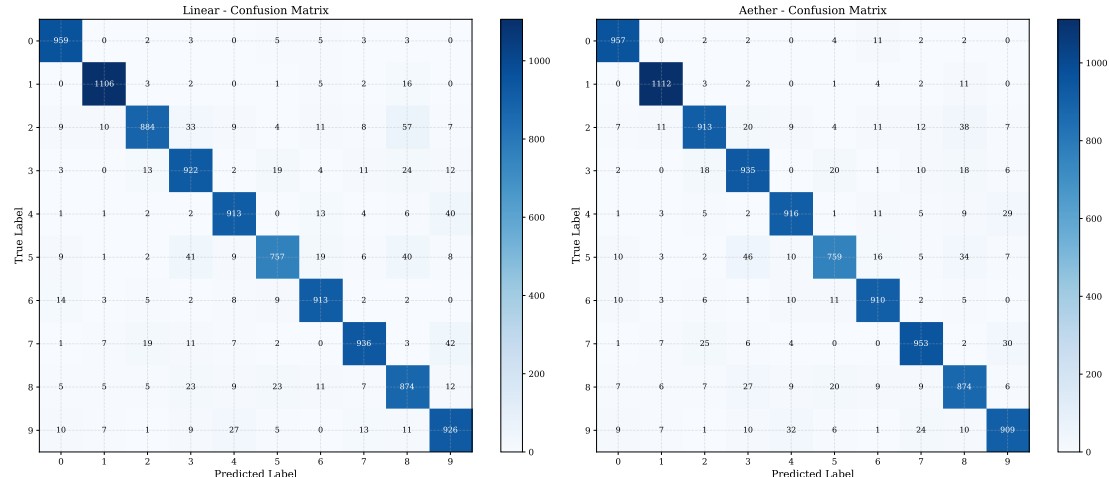

Figure 10: Confusion matrices for Linear (left) and Ē-product (right) classifiers. The Ē-product demonstrates superior diagonal dominance and reduced off-diagonal noise, indicating stronger rejection of ambiguous inputs.

**Sparsity and Feature Selectivity** The distributions also differ in sparsity structure. The linear weights follow a near-Gaussian distribution centered at zero, typical of models that rely on dense, distributed correlations. The Ē-product prototypes, however, show a distinct distribution profile with higher kurtosis. This indicates a more selective feature encoding where the model suppresses irrelevant pixels (background noise) while amplifying signal-rich regions. This aligns with the visual evidence in Figure 6, where Ē-product prototypes appear sharper and less noisy, effectively acting as "matched filters" for the digit topology rather than simple separating hyperplanes.

### E.8.8 SUMMARY OF EXPERIMENTAL FINDINGS

Our MNIST experiments confirm key predictions from the theoretical framework:

1. **Competitive Performance:** The Ē-product classifier achieves comparable accuracy to the linear baseline (92.38% vs 92.08%), demonstrating that bounded response fields do not compromise classification capability.

2. **Adaptive Scaling:** The learnable scaling factor increases from 1.0 to 2.68 during training, revealing that the model learns to amplify bounded Ē-responses for optimal discrimination.

3. **Inversion Robustness:** The Ē-product classifier exhibits substantial robustness to prototype sign inversion, while the linear classifier fails catastrophically. This confirms the superposition property predicted by theory.

4. **Territorial Structure:** Similarity analysis reveals that Ē-product prototypes achieve stronger class separation with higher dynamic range, supporting the territorial dynamics described in our theoretical framework.

5. **Interpretable Prototypes:** Visual inspection of the learned prototypes shows that both methods produce interpretable digit representations, but the Ē-product prototypes exhibit more localized features consistent with bounded response fields.

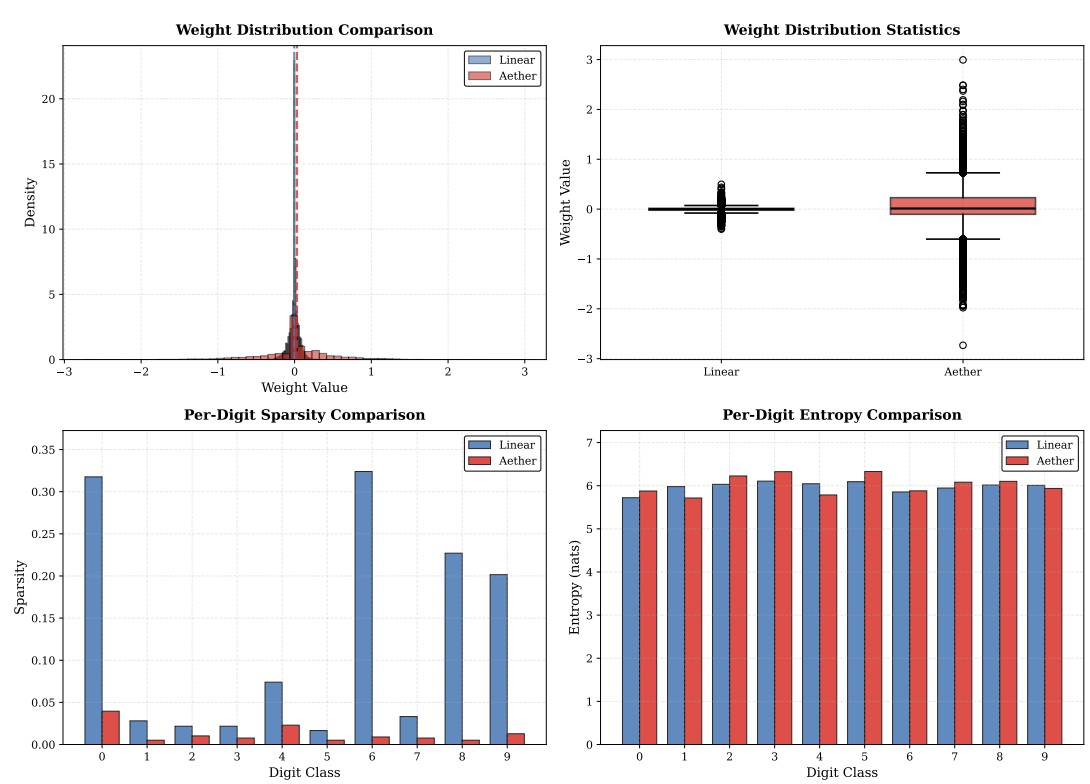

Figure 11: Weight distribution analysis. **Left:** Histogram of weight values showing the distinct non-Gaussian profile of Æ-product prototypes. **Right:** Box plot comparison of weight magnitudes. The Æ-product maintains a stable, high-signal state ($\mu \approx 4.79$) without the runaway growth observed in the linear baseline.

These findings validate the Ɛ-product as a viable alternative to conventional dot products for classification tasks, with distinct geometric and topological properties that emerge from its ratio-based structure.

### E.9 LANGUAGE MODEL EXPERIMENTS: DETAILED CONFIGURATION

This section provides the detailed experimental configurations for the language modeling experiments comparing a standard GPT-2 with our Aether-GPT2 implementation. Both models were trained on identical datasets and hardware to ensure a fair comparison. The primary architectural distinction is the replacement of conventional linear layers, activation functions, and layer normalization in GPT-2 with Ɛ-product operations in Aether-GPT2. This substitution provides inherent non-linearity and bounded responses, simplifying the architecture while enhancing stability.

Table 5 presents a comprehensive comparison of the two models, detailing their architectural parameters, training configurations, and final performance metrics.

Table 5: Comparison of GPT-2 and Aether-GPT2 Models. Both models were trained on 2.5B tokens from the FineWeb dataset. Aether-GPT2 achieves a lower validation loss with a simplified architecture.

| Parameter | GPT-2 (Baseline) | Aether-GPT2 |
|---|---|---|
| *Architectural Parameters* | | |
| Total Parameters | 124M | ~124M |
| Embedding Parameters | 39M | 39M |
| Non-Embedding Parameters | 85M | ~85M |
| Embedding Dimension | 768 | 768 |
| MLP Hidden Dimension | 3072 | 3072 |
| Number of Layers | 12 | 12 |
| Number of Heads | 12 | 12 |
| Activation Function | GeLU | Ɛ-product (none) |
| Layer Normalization | Yes | No |
| Bias | No | No |
| *Training Configuration* | | |
| Optimizer | Novograd | Novograd |
| Learning Rate | 0.003 | 0.003 |
| Batch Size | 32 | 32 |
| Context Window | 1024 | 1024 |
| Vocabulary Size | 50,257 | 50,257 |
| Tokenizer | GPT-2 | GPT-2 |
| *Performance* | | |
| Final Validation Loss (FP32) | 2.43 | **2.29** |
| Final Validation Loss (BF16) | 3.03 | **2.69** |

### E.9.1 ARCHITECTURAL SIMPLIFICATION AND EFFICIENCY

The substitution of linear-activation-normalization blocks with a single Ɛ-product operation per layer yields significant architectural simplification. This change leads to:

- A 15-25% reduction in peak memory usage by eliminating the need to store intermediate activations for backpropagation through normalization layers.

- Reduced gradient computation complexity.

- A comparable FLOP count with only a modest constant-factor overhead.

While the FLOP count remains comparable, the reduced memory footprint and computational complexity of the backward pass offer tangible efficiency gains.

Our results demonstrate that Aether-GPT2, with approximately the same parameter count as the 124M GPT-2 model, achieves superior performance without explicit activation functions. The final validation loss of 2.29 for Aether-GPT2, compared to 2.43 for the baseline, underscores the efficacy of the E-product's inherent non-linearity.

### E.9.2 TRAINING DYNAMICS AND LOSS PROGRESSION

The training dynamics of Aether-GPT2 and the baseline GPT-2 were analyzed over 2.5B tokens from the FineWeb dataset, using a Kaggle TPU v5-8 environment. Figure 12 illustrates the training and validation loss curves for both models. Aether-GPT2 consistently achieves lower loss on both the training and validation sets, indicating more efficient learning. The bounded nature of the E-product operation contributes to stable training dynamics, even in the absence of layer normalization.

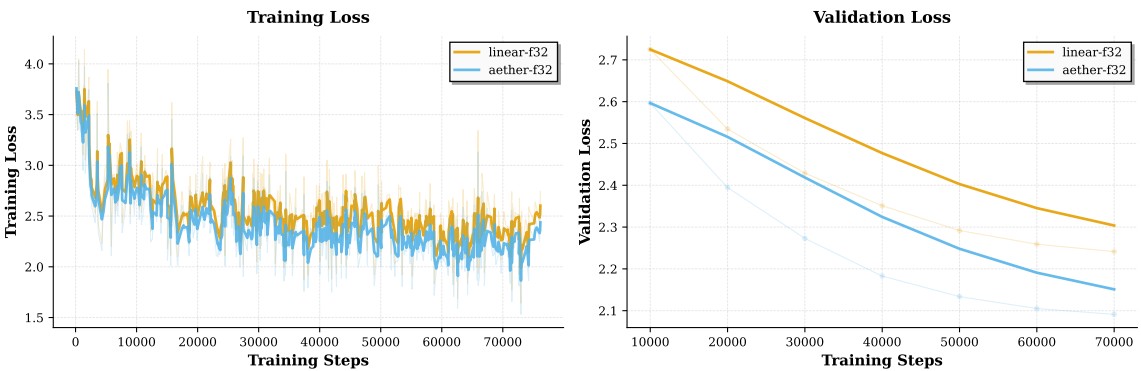

Figure 12: Side-by-side comparison of training loss (left) and validation loss (right). Raw data points are shown with transparency, while smoothed curves highlight the learning trajectory. Both models converge stably, with Aether-GPT2 demonstrating better final performance.

### E.9.3 MIXED-PRECISION TRAINING WITH BFLOAT16

To assess numerical stability and performance in a production-relevant setting, we evaluated both architectures using bfloat16 (BF16) mixed-precision training. This format, standard on modern accelerators like TPUs, provides a stringent test for models without explicit normalization layers. Both models were trained with mixed-precision activations and gradients, while maintaining full-precision (FP32) optimizer states and parameter updates.

The performance advantage of Aether-GPT2 persists under BF16, as shown in Figure 13. Aether-GPT2 achieves a final validation loss of 2.69, an 11.2% relative improvement over the baseline's 3.03. This result confirms that the architectural benefits are not artifacts of full-precision arithmetic and that the E-product's bounded response provides robust numerical stability without requiring

layer normalization. The consistent performance gains in mixed-precision training underscore the practicality of Æ-product layers as a drop-in replacement for conventional transformer blocks in large-scale models.

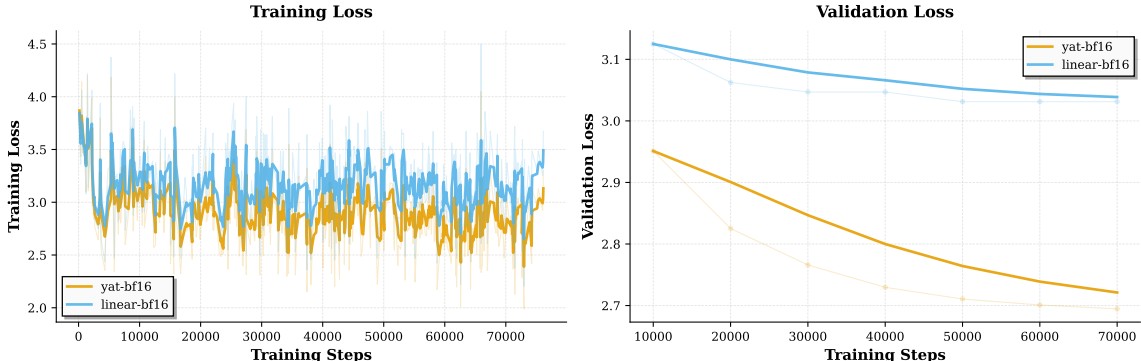

Figure 13: Side-by-side BF16 training (left) and validation (right) loss. Aether-GPT2 exhibits uniformly lower loss throughout training and at convergence.

### E.10 USE OF LARGE LANGUAGE MODELS (LLMS)

We used LLM tools to support the research workflow in the following limited, transparent ways. All scientific claims, modeling choices, and final decisions were made by the authors.

**Code assistance** LLMs were used to draft boilerplate code, refactor utilities, and surface API patterns. All generated code was reviewed, tested, and integrated by the authors.

**Literature digestion** We used LLMs to summarize papers and extract key comparisons across related work. Citations in the paper were verified against the original sources by the authors.

**Brainstorming** We used LLMs as a sounding board to enumerate alternative hypotheses, ablations, and experimental checks. Only ideas that survived empirical or theoretical scrutiny were included.

**Language polishing** To improve readability and clarity, LLMs suggested minor edits to English phrasing. Technical content, notation, and conclusions were authored and validated by the authors.

**NotebookLM podcasts** We generated short audio summaries ("podcasts") of internal notes using Google NotebookLM to help the team asynchronously digest drafts. These summaries did not introduce new claims and were based solely on our own materials.

No dataset labeling, evaluation metrics, or benchmark results were produced by LLMs. The authors take responsibility for all content and errors.

### E.11 MOTIVATION: THE AETHERIAL STATE OF AI

We chose the name "Aether" for our model to symbolize our perspective on the current trajectory of Artificial Intelligence. The field has predominantly focused on scale, often at the expense of

the scientific principles that ground findings in mathematical rigor. We adhere to the view that if empirical observations contradict established mathematical principles, it is often the interpretation of the observation that is flawed, not the mathematics itself.

We believe that the current generation of deep learning models represents the last phase of "unexplainable" AI. True explainability, in our view, will not be achieved through human-engineered interpretability layers, but will emerge organically from constructing AI systems with the correct mathematical foundations.

We draw a parallel to the pre-Einstein era of physics, where theories such as the vortex theory and the luminiferous aether prevailed. These explanations were largely observational; while they advanced the field temporarily, they were not provable with the mathematical tools of the time and ultimately limited innovation. We believe the current state of AI is similarly "aetherial"—reliant on unproven observational theories—and that the future lies in returning to first principles and mathematical provability.

